Proceedings of the 6th Symposium on Advances in Approximate Bayesian Inference, 2024  1–34

# Non-asymptotic approximations of Gaussian neural networks via second-order Poincaré inequalities

**Alberto Bordino**                                      ALBERTO.BORDINO@WARWICK.AC.UK
*University of Warwick*

**Stefano Favaro**                                        STEFANO.FAVARO@EDU.UNITO.IT
*University of Torino and Collegio Carlo Alberto*

**Sandra Fortini**                                        SANDRA.FORTINI@UNIBOCCONI.IT
*Bocconi University*

## Abstract

There is a recent and growing literature on large-width asymptotic and non-asymptotic properties of deep Gaussian neural networks (NNs), namely NNs with weights initialized as Gaussian distributions. For a Gaussian NN of depth $L \geq 1$ and width $n \geq 1$, it is well-known that, as $n \to +\infty$, the NN's output converges (in distribution) to a Gaussian process. Recently, some quantitative versions of this result, also known as quantitative central limit theorems (QCLTs), have been obtained, showing that the rate of convergence is $n^{-1}$, in the 2-Wasserstein distance, and that such a rate is optimal. In this paper, we investigate the use of second-order Poincaré inequalities as an alternative approach to establish QCLTs for the NN's output. Previous approaches consist of a careful analysis of the NN, by combining non-trivial probabilistic tools with ad-hoc techniques that rely on the recursive definition of the network, typically by means of an induction argument over the layers, and it is unclear if and how they still apply to other NN's architectures. Instead, the use of second-order Poincaré inequalities rely only on the fact that the NN is a functional of a Gaussian process, reducing the problem of establishing QCLTs to the algebraic problem of computing the gradient and Hessian of the NN's output, which still applies to other NN's architectures. We show how our approach is effective in establishing QCLTs for the NN's output, though it leads to suboptimal rates of convergence. We argue that such a worsening in the rates is peculiar to second-order Poincaré inequalities, and it should be interpreted as the "cost" for having a straightforward, and general, procedure for obtaining QCLTs.

## 1. Introduction

Let $d, p, L, n \geq 1$ and consider: i) a $d \times p$ matrix (input) $\mathbf{X}$, with $\boldsymbol{x}_j$ being the $j$-th row and $\boldsymbol{x}_u$ being the $u$-th column; ii) an independent random variable (weight) $\mathbf{W} = (\mathbf{W}^{(0)}, \ldots, \mathbf{W}^{(L-1)}, \mathbf{w})$ such that $\mathbf{W}^{(l)} = (w_{i,j_l}^{(l)})$, with the $w_{i,j_l}^{(l)}$ 's being i.i.d. as Gaussian $\mathcal{N}(0, \sigma_w^2)$ for $l = 0, \ldots, L-1$, $1 \leq i \leq n, 1 \leq j_0 \leq d, 1 \leq j_l \leq n$, and $\mathbf{w} = (w_1, \ldots, w_n)$, with the $w_i$'s being i.i.d as Gaussian $\mathcal{N}(0, \sigma_w^2)$ for $i = 1, \ldots, n$; ii) an independent random variable (bias) $\mathbf{b} = (\mathbf{b}^{(0)}, \ldots, \mathbf{b}^{(L-1)}, b)$ such that $\mathbf{b}^{(l)} = (b_1^{(l)}, \ldots, b_n^{(l)})$, with the $b_i^{(l)}$ 's being i.i.d. as $\mathcal{N}\left(0, \sigma_b^2\right)$ for $l = 0, \ldots, L-1$ and $i = 1, \ldots, n$, and with $b$ being distributed

as $\mathcal{N}\left(0, \sigma_b^2\right)$. For a function $\tau : \mathbb{R} \to \mathbb{R}$ (activation), a fully-connected feed-forward deep Gaussian neural network (NN) of depth $L$ and width $n$ is defined as

$$
\begin{cases}
f_i^{(1)}(\mathbf{X}) = \sum_{j=1}^d w_{i,j}^{(0)} \mathbf{x}_j + b_i^{(0)} \mathbf{1}^T \\[2mm]
f_i^{(l)}(\mathbf{X}, n) = \frac{1}{\sqrt{n}} \sum_{j=1}^n w_{i,j}^{(l-1)}(\tau \circ f_j^{(l-1)}(\mathbf{X}, n)) + b_i^{(l-1)} \mathbf{1}^T \qquad l = 2, \ldots, L \\[2mm]
f^{(L+1)}(\mathbf{X}, n) = b + \frac{1}{\sqrt{n}} \sum_{i=1}^n w_i \tau(f_i^{(L)}(\mathbf{X}, n)),
\end{cases} \tag{1}
$$

with $\mathbf{1}$ and $\circ$ denoting the $p$ dimensional column vector of 1's and the element-wise application, respectively. Let $(f_i^{(l)}(\mathbf{X}, n))_{i \geq 1}$ be the sequence or random variables that is obtained by extending $(\mathbf{W}^{(0)}, \ldots, \mathbf{W}^{(L-1)})$ and $(\mathbf{b}^{(0)}, \ldots, \mathbf{b}^{(L-1)})$ to infinite independent arrays, for $l = 0, \ldots, L-1$. Under the assumption that $\tau$ is continuous and such that $|\tau(s)| \leq \alpha + \beta|s|$ for every $s \in \mathbb{R}$ and some $\alpha, \beta \geq 0$, (Matthews et al., 2018, Theorem 4) shows that as $n \to +\infty$ jointly over the first $l \geq 1$ NN's layers

$$
(f_i^{(l)}(\mathbf{X}, n))_{i \geq 1} \xrightarrow{w} (f_i^{(l)}(\mathbf{X}))_{i \geq 1}, \tag{2}
$$

where $(f_i^{(l)}(\mathbf{X}))_{i \geq 1}$, as a stochastic process indexed by $\mathbf{X}$, is distributed according to the product measure of $p$-dimensional Gaussian measures, namely $\otimes_{i \geq 1} \mathcal{N}_p(\mathbf{0}, \Sigma^{(l)})$, for a suitable specification of $\Sigma^{(l)}$. The work of Matthews et al. (2018) improves over previous results of Neal (1996) and Lee et al. (2018b), and it has been later refined and generalized in, e.g., Garriga-Alonso et al. (2018), Lee et al. (2018a), Novak et al. (2018), Antognini (2019), Yang (2019), Aitken and Gur-Ari (2020), Andreassen and Dyer (2020), Bracale et al. (2021), Favaro et al. (2022), Lee et al. (2024) and Hanin (2023).

There is a recent interest in quantitative versions of (2), also known as quantitative central limit theorems (QCLTs), characterizing the rate of convergence of the NN's output $f^{(L+1)}(\mathbf{X}, n)$ to its infinite-wide limit, with respect to suitable distances. This problem was first investigated by Eldan et al. (2021) for a shallow NN, i.e. $L = 1$, on the $(d-1)$-sphere with Gaussian distributed $w_{i,j}$'s and Rademacher distributed $w_i$'s. In particular, assuming a polynomial activation function, Eldan et al. (2021) established a functional QCLT in the 2-Wasserstein distance $d_{W_2}$. Some refinements of the work of Eldan et al. (2021), still for shallow NNs on the $(d-1)$-sphere, are in Klukowski (2022), assuming the $w_{i,j}$'s to be Uniformly distributed and the $w_i$'s to have a general distribution with finite fourth moment, and in Cammarota et al. (2023), assuming the $w_{i,j}$'s to be Gaussian distributed. In the more general setting of deep NNs, i.e. $L \geq 2$, Basteri and Trevisan (2022) established a QCLT in $d_{W_2}$ for the NN's output $f^{(L+1)}(\mathbf{X}, n)$. In particular, if $N \sim \mathcal{N}(\mathbf{0}, \Sigma^{(L+1)})$ is the infinite-wide limit of $f^{(L+1)}(\mathbf{X}, n)$, then, assuming a Lipschitz activation function $\tau$, Basteri and Trevisan (2022) proved that

$$
d_{W_2}(f^{(L+1)}(\mathbf{X}, n), N) \lesssim n^{-1/2} \tag{3}
$$

Apollonio et al. (2023) and Favaro et al. (2023) generalized the result of Basteri and Trevisan (2022) to a broader class of activation functions and to general convex distances. Denoting by $d_{W_1}$ and $d_{TV}$ the 1-Wasserstein distance and the total variation distance, respectively,

for $p = 1$ Favaro et al. (2023) proved that

$$\min \left\{ d_{W_1}(f^{(L+1)}(\mathbf{X}, n), N), d_{TV}(f^{(L+1)}(\mathbf{X}, n), N) \right\} \gtrsim n^{-1} \tag{4}$$

and established a corresponding upper bound that matches the order $n^{-1}$. Interestingly, and somehow surprisingly, this result shows that for $p = 1$ the optimal rate of convergence is of order $n^{-1}$. In Trevisan (2023), an analogous upper bound of order $n^{-1}$ is achieved for a number $p \geq 1$ of inputs. We refer to Balasubramanian et al. (2024) for QCLTs in the context of NNs with non-Gaussian weights.

## 1.1. Our contributions

In this paper, we investigate the use of second-order Poincaré inequalities to establish QCLTs for $f^{(L+1)}(\mathbf{X}, n)$, providing an alternative approach to those developed in Basteri and Trevisan (2022), Apollonio et al. (2023), Favaro et al. (2023) and Trevisan (2023). Second-order Poincaré inequalities were introduced in Chatterjee (2009) and Nourdin et al. (2009) as a tool to obtain QCLTs for functionals of Gaussian processes, estimating the approximation error between the functional of interest and a Gaussian process, with respect to suitable distances. By using the fact that Gaussian NNs are functionals of Gaussian processes, we establish QCLTs for $f^{(L+1)}(\mathbf{X}, n)$ by means of a direct application of second-order Poincaré inequalities available in the literature. In particular, we make use of some recent refinements of second-order Poincaré inequalities introduced by Vidotto (2020), which provide tight estimates of the approximation error. For $L = 1$, assuming $\tau \in C^2(\mathbb{R})$ such $\tau$ and its first and second derivatives are bounded above by $\alpha + \beta|x|^\gamma$, for $\alpha, \beta, \gamma \geq 0$, we show that

$$d_{W_1}(f^{(2)}(\mathbf{X}, n), N) \lesssim n^{-1/2}. \tag{5}$$

Then, we consider the more general setting $L \geq 1$. For $L = 2$, under analogous assumptions on $\tau$, we show that

$$d_{W_1}(f^{(3)}(\mathbf{X}, n), N) \lesssim n^{-1/4}, \tag{6}$$

and conjecture the same rate of convergence for any $L \geq 2$. Both (5) and (6) follow from a direct application of results in Vidotto (2020), which require the sole computation of the gradient and Hessian of the NN's output. In other terms, for any $L \geq 1$, the use of second-order Poincaré inequalities reduces the problem of establishing QCLTs for $f^{(L+1)}(\mathbf{X}, n)$ to the algebraic problem of computing some derivatives of the NN's output. While it becomes unwieldy as $L$ increases, the computation of the gradient and the Hessian is standard, leading to explicit expressions. In particular, for $L = 1$ our approach provides a straightforward proof of a QCLTs for $f^{(L+1)}(\mathbf{X}, n)$, with rate $n^{-1/2}$.

Our analysis shows how second-order Poincaré inequalities are an effective tool to obtain QCLTs for $f^{(L+1)}(\mathbf{X}, n)$, though they lead to suboptimal rates of convergence with respect to previous works: i) for $L = 1$, i.e. (5), we achieve the same rate $n^{-1/2}$ as in Basteri and Trevisan (2022) and Apollonio et al. (2023), which, however, is known to be suboptimal from Favaro et al. (2023) and Trevisan (2023); ii) for $L \geq 2$, i.e. (6), we obtain the rate $n^{-1/4}$, while Basteri and Trevisan (2022) and Apollonio et al. (2023) still achieve the rate $n^{-1/2}$, as well as Favaro et al. (2023) and Trevisan (2023) still achieve the rate $n^{-1}$. We argue that such a worsening in the rate of convergence is peculiar to second-order

Poincaré inequalities, and it should be interpreted as the "cost" for having a straightforward, and general, procedure to obtain QCLTs for $f^{(L+1)}(\mathbf{X}, n)$. The approaches of Basteri and Trevisan (2022), Apollonio et al. (2023), Favaro et al. (2023) and Trevisan (2023) consist of a careful analysis of $f^{(L+1)}(\mathbf{X}, n)$, by combining non-trivial probabilistic tools with ad-hoc techniques that rely on the recursive definition (1), typically by means of an induction argument over the NN's layers; as such, it is unclear if and how these approaches still apply to other NN's architectures, such as convolutional NNs and generalizations thereof. Instead, our approach consists of a direct application of the results of Vidotto (2020) to $f^{(L+1)}(\mathbf{X}, n)$, which, by relying only on the fact that $f^{(L+1)}(\mathbf{X}, n)$ is a functional of a Gaussian process, it applies to a broad range of NN's architectures, still reducing the problem of establishing QCLTs to the algebraic problem of computing gradients and Hessians.

### 1.2. Organization of the paper

The paper is structured as follows. In Section 2 we present an overview on second-order Poincaré inequalities, recalling some results of Vidotto (2020). Section 3 contains our results for shallow Gaussian NNs, whereas in Section 4 we extend these results to deep Gaussian NNs. In Section 5 we discuss our results and directions of future research. Proofs and numerical illustrations are in the appendix.

## 2. Preliminaries on second-order Poincaré inequalities

We denote by $(\Omega, \mathcal{F}, \mathbb{P})$ the probability space on which random variables are assumed to be defined, and by $\|X\|_{L^q} := (\mathbb{E}[X^q])^{1/q}$ the $L^q$ norm of a random variable $X$. We consider some popular distances between (probability) distributions of real-valued random variables. In particular, let $X$ and $Y$ be two random variables in $\mathbb{R}^d$, for some $d \geq 1$. We denote by $d_{W_1}$ the 1-Wasserstein distance, i.e.,

$$d_{W_1}(X, Y) = \sup_{h \in \mathscr{H}} |\mathbb{E}[h(X)] - \mathbb{E}[h(Y)]|,$$

where $\mathscr{H}$ is the class of all functions $h : \mathbb{R}^d \to \mathbb{R}$ such that it holds true that $\|h\|_{\mathrm{Lip}} \leq 1$, with $\|h\|_{\mathrm{Lip}} = \sup_{x,y \in \mathbb{R}^d, x \neq y} |h(x) - h(y)|/\|x - y\|_{\mathbb{R}^d}$. Further, let $d_{TV}$ be the total variation distance, i.e.,

$$d_{TV}(X, Y) = \sup_{B \in \mathscr{B}(\mathbb{R}^m)} |\mathbb{P}(X \in B) - \mathbb{P}(Y \in B)|,$$

where $\mathscr{B}(\mathbb{R}^d)$ is the Borel $\sigma$-field of $\mathbb{R}^d$. Finally, let $d_{KS}$ be the Kolmogorov-Smirnov distance, i.e.,

$$d_{KS}(X, Y) = \sup_{z_1, \ldots, z_d \in \mathbb{R}} |\mathbb{P}\left(X \in \times_{i=1}^d (-\infty, z_i]\right) - \mathbb{P}\left(Y \in \times_{i=1}^d (-\infty, z_i]\right)|.$$

In particular, it is useful to recall that: i) $d_{KS}(\cdot, \cdot) \leq d_{TV}(\cdot, \cdot)$; ii) if $X$ is a real-valued random variable and $N \sim \mathcal{N}(0, 1)$ is the standard Gaussian random variable then $d_{KS}(X, N) \leq 2\sqrt{d_{W_1}(X, N)}$.

Second-order Poincaré inequalities provide a well-known tool to obtain Gaussian approximations of functionals of Gaussian fields, with respect to suitable distances (Chatterjee (2009); Nourdin et al. (2009)). See also Nourdin and Peccati (2012) for details. If

$N \sim \mathcal{N}(0,1)$ then the Gaussian PI states that

$$\text{Var}[f(N)] \leq \mathbb{E}[f'(N)^2] \tag{7}$$

for every differentiable function $f : \mathbb{R} \to \mathbb{R}$, a result that was first discovered in the seminal work of Nash (1956), and then reproved by Chernoff (1981). The inequality (7) implies that if the $L^2$ norm of the random variable $f'(N)$ is small, then so are the fluctuations of the random variable $f(N)$. The first version of a second-order Poincaré inequality was obtained in Chatterjee (2009), where it is proved that one can iterate (7) in order to assess the total variation distance between the distribution of $f(N)$ and the distribution of a Gaussian random variable with matching mean and variance.

**Theorem 1 (Chatterjee (2009))** *For any $d \geq 1$, let $X \sim \mathcal{N}(0, I_{d \times d})$. Consider $f \in C^2(\mathbb{R}^d)$ such that $\nabla f$ and $\nabla^2 f$, and denote the gradient of $f$ and Hessian of $f$, respectively. Further, suppose that $f(X)$ has a finite fourth moment, and let $\mu = \mathbb{E}[f(X)]$ and $\sigma^2 = \text{Var}[f(X)]$. If $N \sim \mathcal{N}(\mu, \sigma^2)$ then*

$$d_{TV}(f(X), N) \leq \frac{2\sqrt{5}}{\sigma^2} \left\{ \mathbb{E}\left[ \|\nabla f(X)\|_{\mathbb{R}^d}^4 \right] \right\}^{1/4} \left\{ \mathbb{E}\left[ \|\nabla^2 f(X)\|_2^4 \right] \right\}^{1/4}, \tag{8}$$

*where $\|\cdot\|_2$ stands for the operator norm of the Hessian $\nabla^2 f(X)$ regarded as a random $d \times d$ matrix.*

By combining Stein's method and Malliavin calculus, Nourdin et al. (2009) obtained a more general version of (8), involving functionals of arbitrary infinite-dimensional Gaussian fields. Both (8) and its generalization in Nourdin et al. (2009) are known to provide estimates of the approximations error that are not tight. This is because, in general, it is not possible to compute explicitly the expected value of the operator norm involved in the estimate of total variation distance, which leads to move further away from the distance in distribution, and further bound the operator norm. To overcome this drawback, the work of Vidotto (2020) adapted to the Gaussian setting an approach recently developed in Last et al. (2016) to obtain second-order Poincaré inequalities for Gaussian approximation of Poisson functionals, which yields to estimates of the approximation error that are tight.

**Theorem 2 (Vidotto (2020) - 1-dimensional case)** *For any $d \geq 1$, let $X \sim \mathcal{N}(0, I_{d \times d})$. Consider $f \in C^2(\mathbb{R}^d)$ such that the partial derivatives of the function $f$ have sub-exponential growth, let $F = f(X)$ such that $E[F] = 0$ and $E\left[F^2\right] = \sigma^2$, and denote by $\nabla_i F$ and $\nabla_{i,\cdot}^2 F$ the $i$-th element of the gradient of $F$ and the $i$-th row of the Hessian of $F$, respectively. If $N \sim \mathcal{N}\left(0, \sigma^2\right)$ then*

$$d_M(F, N) \leq c_M \sqrt{\sum_{l,m=1}^d \left\{ \mathbb{E}\left[ \left( \langle \nabla_{l,\cdot}^2 F, \nabla_{m,\cdot}^2 F \rangle \right)^2 \right] \right\}^{1/2} \left\{ \mathbb{E}\left[ (\nabla_l F \nabla_m F)^2 \right] \right\}^{1/2}}, \tag{9}$$

*where $\langle \cdot, \cdot \rangle$ denotes the scalar product, $M \in \{TV, KS, W_1\}$, $c_{TV} = \frac{4}{\sigma^2}$, $c_{KS} = \frac{2}{\sigma^2}$ and $c_{W_1} = \sqrt{\frac{8}{\sigma^2 \pi}}$.*

The next theorem generalizes Theorem 2 to multidimensional ($p \geq 1$) functionals of Gaussian random variables. We refer to Appendix A for a more detailed overview of the results of Vidotto (2020)

**Theorem 3 (Vidotto (2020) - $p$-dimensional case)** *For any $d \geq 1$, let $X \sim \mathcal{N}(0, I_{d \times d})$. For any $p \geq 1$ consider $f_1, \ldots, f_p \in C^2(\mathbb{R}^d)$ such that the partial derivatives of $f_i$ have subexponential growth, for $i = 1, \ldots, p$, let $[F_1 \ \ldots \ F_p] = [f_1(X) \ \ldots \ f_p(X)]$ such that $E[F_i] = 0$ for $i = 1, \ldots, p$ and $E[F_i F_j] = c_{ij}$ for $i, j = 1, \ldots, p$, with $C = \{c_{ij}\}_{i,j=1,\ldots,p}$ being a symmetric and positive definite matrix, i.e. a variance-covariance matrix, and denote by $\nabla_i F$ and $\nabla^2_{i,.} F$ the $i$-th element of the gradient of $F$ and the $i$-th row of the Hessian of $F$, respectively. If $N \sim \mathcal{N}(0, C)$, then*

$$d_{W_1}(F, N) \leq 2\sqrt{p} \left\| C^{-1} \right\|_2 \left\| C \right\|_2$$
$$\times \sqrt{\sum_{i,k=1}^{p} \sum_{l,m=1}^{d} \left\{ \mathbb{E}\left[\left(\langle \nabla^2_{l,.} F_i, \nabla^2_{m,.} F_i \rangle\right)^2\right]\right\}^{1/2} \left\{ \mathbb{E}\left[(\nabla_l F_k \nabla_m F_k)^2\right]\right\}^{1/2}} \quad (10)$$

*where $\|\cdot\|_2$ is the spectral norm of a matrix.*

## 3. QCLTs for shallow NNs

We make use of Theorem 2 and Theorem 3 to obtain QCLTs for $F = f^{L+1}(X, n)$, with $f^{L+1}(X, n)$ defined in (1), with $L = 1$. We start with a 1-dimensional unitary input, i.e. $d = 1$ and $x = 1$, unit variance's weight, i.e. $\sigma_w^2 = 1$, and no biases, i.e. $b_i^{(0)} = b = 0$ for any $i \geq 1$. That is, we consider

$$F = \frac{1}{n^{1/2}} \sum_{j=1}^{n} w_j \tau(w_j^{(0)}). \quad (11)$$

By a straightforward calculation, $\mathbb{E}[F] = 0$ and $\text{Var}[F] = \mathbb{E}_{Z \sim \mathcal{N}(0,1)}[\tau^2(Z)]$. Since $F$ is a function of independent standard Gaussian random variables, Theorem 2 can be applied to approximate $F$ with a Gaussian random variable with the same mean and variance as $F$, quantifying the approximation error.

**Theorem 4** *Let $F$ in (11) with $\tau \in C^2(\mathbb{R})$ such that $|\tau(x)| \leq \alpha + \beta|x|^\gamma$ and $\left|\frac{d^l}{dx^l}\tau(x)\right| \leq \alpha + \beta|x|^\gamma$ for $l = 1, 2$ and some $\alpha, \beta, \gamma \geq 0$. If $N \sim \mathcal{N}(0, \sigma^2)$ with $\sigma^2 = \mathbb{E}_{Z \sim \mathcal{N}(0,1)}[\tau^2(Z)]$, then for any $n \geq 1$*

$$d_M(F, N) \leq \frac{c_M}{\sqrt{n}} \sqrt{3(1 + \sqrt{2})} \cdot \left\| \alpha + \beta|Z|^\gamma \right\|_{L_4}^2, \quad (12)$$

*where $Z \sim \mathcal{N}(0, 1)$, $M \in \{TV, KS, W_1\}$, with constants $c_{TV} = 4/\sigma^2$, $c_{KS} = 2/\sigma^2$, and $c_{W_1} = \sqrt{8/\sigma^2\pi}$.*

See Appendix B for the proof of Theorem 4. The proof follows by an application of Theorem 2, reducing the problem of establishing the QCLT to the algebraic problem of

computing the gradient and the Hessian of the NN. The QCLT (12) has the convergence rate $n^{-1/2}$ with respect to the 1-Wasserstein distance, the total variation distance and the Kolmogorov-Smirnov distance. The rate $n^{-1/2}$ is also obtained in Basteri and Trevisan (2022) and Apollonio et al. (2023), through different techniques, whereas Favaro et al. (2023) proved a QCLT with rate $n^{-1}$, also proving the optimality of such a rate. As for the constant in (12), it depends on $\mathbb{E}_{Z\sim\mathcal{N}(0,1)}[\tau^2(Z)]$, which can be evaluated exactly or approximated once $\tau$ is specified. Theorem 4 can be extended to an input $\boldsymbol{x} \in \mathbb{R}^d$, showing that the problem of QCLT still reduces to the application of Theorem 2. In particular, we can write

$$F := \frac{1}{n^{1/2}}\sigma_w \sum_{j=1}^{n} w_j \tau(\sigma_w \langle w_j^{(0)}, \boldsymbol{x}\rangle + \sigma_b b_j^{(0)}) + \sigma_b b, \tag{13}$$

with $w_j^{(0)} = [w_{j,1}^{(0)}, \ldots, w_{j,d}^{(0)}]^T$ and $w_j \overset{d}{=} w_{j,i}^{(0)} \overset{iid}{\sim} \mathcal{N}(0,1)$. We set $\Gamma^2 = \sigma_w^2\|\boldsymbol{x}\|^2 + \sigma_b^2$, and for $n \geq 1$ we consider a collection $(Y_1, \ldots, Y_n)$ of independent standard Gaussian random variables. Then, from (13)

$$F \overset{d}{=} \frac{1}{n^{1/2}}\sigma_w \sum_{j=1}^{n} w_j \tau(\Gamma Y_j) + \sigma_b b.$$

As before, $\mathbb{E}[F] = 0$ and $\text{Var}[F] = \sigma_w^2 \mathbb{E}_{Z\sim\mathcal{N}(0,1)}[\tau^2(\Gamma Z)] + \sigma_b^2$. Since $F$ in (13) is a function of independent standard Gaussian random variables, Theorem 2 can be applied to approximate $F$ with a Gaussian random variable with the same mean and variance as $F$, quantifying the approximation error.

**Theorem 5** *Let $F$ in (13) with $\tau \in C^2(\mathbb{R})$ such that $|\tau(x)| \leq \alpha + \beta|x|^\gamma$ and $\left|\frac{d^l}{dx^l}\tau(x)\right| \leq \alpha + \beta|x|^\gamma$ for $l = 1, 2$ and some $\alpha, \beta, \gamma \geq 0$. If $N \sim \mathcal{N}(0, \sigma^2)$ with $\sigma^2 = \sigma_w^2 \mathbb{E}_{Z\sim\mathcal{N}(0,1)}[\tau^2(\Gamma Z)] + \sigma_b^2$ and $\Gamma = (\sigma_w^2\|\boldsymbol{x}\|^2 + \sigma_b^2)^{1/2}$, then for any $n \geq 1$*

$$d_M(F, N) \leq \frac{c_M \sqrt{\Gamma^2 + \Gamma^4(2 + \sqrt{3(1 + 2\Gamma^2 + 3\Gamma^4)})}\|\alpha + \beta|\Gamma Z|^\gamma\|_{L^4}^2}{\sqrt{n}}, \tag{14}$$

*where $Z \sim \mathcal{N}(0,1)$, $M \in \{TV, KS, W_1\}$, with constants $c_{TV} = 4/\sigma^2, c_{KS} = 2/\sigma^2, c_{W_1} = \sqrt{8/\sigma^2\pi}$.*

See Appendix C for the proof of Theorem 5. We conclude by extending Theorem 5 to $p > 1$ inputs $(\boldsymbol{x_1}, \ldots, \boldsymbol{x_p})$, where $\boldsymbol{x_i} \in \mathbb{R}^d$ for $i = 1, \ldots, p$. In particular, we consider $F = [F_1 \ \ldots \ F_p]$, where

$$F_i := \frac{1}{n^{1/2}}\sigma_w \sum_{j=1}^{n} w_j \tau(\sigma_w \langle w_j^{(0)}, \boldsymbol{x_i}\rangle + \sigma_b b_j^{(0)}) + \sigma_b b, \tag{15}$$

with $w_j^{(0)} = [w_{j,1}^{(0)}, \ldots, w_{j,d}^{(0)}]^T$ and $w_j \overset{d}{=} w_{j,i}^{(0)} \overset{d}{=} b_j^{(0)} \overset{d}{=} b \overset{iid}{\sim} \mathcal{N}(0,1)$. Under this setting, Theorem 3 can be applied to obtain an approximation of $F$ with a Gaussian random vector

whose mean and covariance are the same as $F$. The resulting estimate depends on the maximum and the minimum eigenvalues, i.e. $\lambda_1(C)$ and $\lambda_p(C)$ respectively, of the covariance matrix $C$, whose $(i, k)$-th entry is given by

$$\mathbb{E}[F_i F_k] = \sigma_w^2 \mathbb{E}[\tau(Y_i)\tau(Y_k)] + \sigma_b^2, \tag{16}$$

where

$$Y \sim \mathcal{N}(0, \sigma_w^2 \mathbf{X}^T \mathbf{X} + \sigma_b^2 \mathbf{1}\mathbf{1}^T)$$

with $\mathbf{1}$ being the all-one vector of dimension $p$, and with $\mathbf{X}$ being the $n \times p$ matrix of the inputs $\{\boldsymbol{x_i}\}_{i \in [p]}$.

**Theorem 6** *Let $F = [F_1 \ \ldots \ F_p]$ with $F_i$ being the NN output in (15), for $i = 1, \ldots, p$, with $\tau \in C^2(\mathbb{R})$ such that $|\tau(x)| \leq \alpha + \beta|x|^\gamma$ and $\left|\frac{d^l}{dx^l}\tau(x)\right| \leq \alpha + \beta|x|^\gamma$ for $l = 1, 2$ and some $\alpha, \beta, \gamma \geq 0$. Furthermore, let $C$ be the covariance matrix of $F$, whose entries are given in (16), and define $\Gamma_i^2 = \sigma_w^2\|\boldsymbol{x_i}\|^2 + \sigma_b^2$ and $\Gamma_{ik} = \sigma_w^2 \sum_{j=1}^d |x_{ij}x_{kj}| + \sigma_b^2$. If $N = [N_1 \ \cdots \ N_p] \sim \mathcal{N}(0, C)$, then for any $n \geq 1$*

$$d_{W_1}(F, N) \leq 2\sigma_w^2 \tilde{K} \frac{\lambda_1(C)}{\lambda_p(C)} \sqrt{\frac{p}{n}}, \tag{17}$$

*where $\lambda_1(C)$ and $\lambda_p(C)$ are the maximum and the minimum eigenvalues of $C$, respectively, and where*

$$\tilde{K} = \left\{ \sum_{i,k=1}^p (\Gamma_i^2 + \sqrt{3(1 + 2\Gamma_i^2 + 3\Gamma_i^4)}\Gamma_{ik}^2 + 2\Gamma_i^2\Gamma_{ik})\|\alpha + \beta|\Gamma_i Z|^\gamma\|_{L^4}^2\|\alpha + \beta|\Gamma_k Z|^\gamma\|_{L^4}^2 \right\}^{1/2},$$

*with $Z \sim \mathcal{N}(0, 1)$.*

See Appendix D for the proof of Theorem 6. Along the same lines of the proofs of Theorem 4 and Theorem 5, Theorem 6 follows by a direct application of Theorem 3, which boils down to straightforward (algebraic) calculations for the gradient and the Hessian of the NN. The estimate (17) of the approximation error $d_{W_1}(F, N)$ has the expected convergence rate $n^{-1/2}$ with respect to the 1-Wasserstein distance, with a constant depending on the spectral norms of the covariance matrix $C$ and the precision matrix $C^{-1}$. In particular, such spectral norms must be computed explicitly for the specific activation $\tau$ in use, or at least bounded from above, in order to apply Theorem 6. This boils down to finding the greatest eigenvalue $\lambda_1$ and the smallest eigenvalue $\lambda_p$ of the matrix $C$, which can be done for a broad class of activations with classical optimization techniques, or at least bounding $\lambda_1$ from above and $\lambda_p$ from below (Diaconis and Stroock (1991); Guattery et al. (1999)). Within the setting of Theorem 6, the rate $n^{-1/2}$ is also obtained in Basteri and Trevisan (2022) and Apollonio et al. (2023), through different techniques, whereas Trevisan (2023) proved a QCLT with rate $n^{-1}$.

## 4. QCLTs for deep NNs

We consider the use of the second-order Poincaré inequalities to obtain QCLTs for the output of a deep Gaussian NN, generalizing Theorem 6. For $L \geq 2$ second-order Poincaré inequalities may be applied to the NN's output $f^{(L+1)}(\mathbf{X}, n)$ along the same lines as for $L = 1$. In particular, the use of second-order Poincaré inequalities still relies on the computation of the gradient and the Hessian of the NN's output, which for $L \geq 2$ is a non-trivial task due to its (algebraic) complexity that increases with the depth $L$ of the NN. Let $F := f^{(L+1)}(\mathbf{X}, n)$, with $f^{(L+1)}(\mathbf{X}, n)$ as in (1). Since $F$ is a function of independent Gaussian random variables, Theorem 3 can be applied to give an upper bound for the 1-Wasserstein distance between $F$ and a Gaussian random vector with the same covariance matrix. See Appendix E for explicit expressions of the gradient and the Hessian of the NN output.

**Theorem 7** *Let $F = [F_1 \ \dots \ F_p] := f^{(L+1)}(\mathbf{X}, n)$ with $f^{(L+1)}(\mathbf{X}, n)$ being the output of the NN defined in (1), and let $C$ be the covariance matrix of $F$. If $N = [N_1 \ \cdots \ N_p] \sim \mathcal{N}(0, C)$, then for any $n \geq 1$ and $L > 1$*

$$d_{W_1}(F, N) \leq 2\sqrt{p}\frac{\lambda_1(C)}{\lambda_p(C)}$$

$$\times \left\{ \sum_{i,k=1}^{p} \sum_{l,m=0}^{L-1} \sum_{i_1,i_2,i_3,i_4=1}^{n} \left\{ \mathbb{E}\left[\left(\left\langle \nabla^2_{w^{(l)}_{i_1,i_2},\cdot} F_i, \nabla^2_{w^{(m)}_{i_3,i_4},\cdot} F_k \right\rangle\right)^2\right] \mathbb{E}\left[\left(\frac{\partial F_i}{\partial w^{(l)}_{i_1,i_2}} \frac{\partial F_k}{\partial w^{(m)}_{i_3,i_4}}\right)^2\right]\right\}^{1/2}\right.$$

$$+ 2 \sum_{i,k=1}^{p} \sum_{l=0}^{L-1} \sum_{i_1,i_3,i_4=1}^{n} \left\{ \mathbb{E}\left[\left(\left\langle \nabla^2_{w_{i_1},\cdot} F_i, \nabla^2_{w^{(l)}_{i_3,i_4},\cdot} F_k \right\rangle\right)^2\right] \mathbb{E}\left[\left(\frac{\partial F_i}{\partial w_{i_1}} \frac{\partial F_k}{\partial w^{(l)}_{i_3,i_4}}\right)^2\right]\right\}^{1/2}$$

$$+ \left. \sum_{i,k=1}^{p} \sum_{i_1,i_3=1}^{n} \left\{ \mathbb{E}\left[\left(\left\langle \nabla^2_{w_{i_1},\cdot} F_i, \nabla^2_{w_{i_3},\cdot} F_k \right\rangle\right)^2\right] \mathbb{E}\left[\left(\frac{\partial F_i}{\partial w_{i_1}} \frac{\partial F_k}{\partial w_{i_3}}\right)^2\right]\right\}^{1/2}\right\}^{1/2}.$$

The estimate of $d_{W_1}(F, N)$ in Theorem 7 is implicit, as controlling the expectations involving the gradient and the Hessian of the NN is a non-trivial task for a general depth $L \geq 2$. If $p = 1$, with $\mathbf{X} = \boldsymbol{x}$, then

$$\mathbb{E}\left[\left(\frac{\partial F}{\partial w_i} \frac{\partial F}{\partial w_j}\right)^2\right] = \left(\frac{\sigma_w}{\sqrt{n}}\right)^4 \mathbb{E}\left[\tau\left(f^{(L)}_i(\boldsymbol{x}, n)\right)^2 \tau\left(f^{(L)}_j(\boldsymbol{x}, n)\right)^2\right]. \tag{18}$$

As the random variables on the right-hand side of (18) are dependent, to deal with the expectation one may consider to condition with respect to the output of the previous layer, and then make use the fact that $f^{(L)}_i(\boldsymbol{x}, n)$ and $f^{(L)}_j(\boldsymbol{x}, n)$ are conditionally i.i.d. given $f^{(L-1)}_.(\boldsymbol{x}, n)$. Then, (18) factorizes as

$$\mathbb{E}\left[\left(\frac{\partial F}{\partial w_i} \frac{\partial F}{\partial w_j}\right)^2\right]$$

$$= \left(\frac{\sigma_w}{\sqrt{n}}\right)^4 \mathbb{E}\left[\tau\left(f_i^{(L)}(\boldsymbol{x},n)\right)^2 \tau\left(f_j^{(L)}(\boldsymbol{x},n)\right)^2\right]$$

$$= \left(\frac{\sigma_w}{\sqrt{n}}\right)^4 \mathbb{E}\left[\mathbb{E}\left[\tau\left(f_i^{(L)}(\boldsymbol{x},n)\right)^2 \tau\left(f_j^{(L)}(\boldsymbol{x},n)\right)^2 \bigg| f_{\cdot}^{(L-1)}(\boldsymbol{x},n)\right]\right]$$

$$\stackrel{\text{cond. i.i.d.}}{=} \left(\frac{\sigma_w}{\sqrt{n}}\right)^4 \mathbb{E}\left[\mathbb{E}\left[\tau\left(f_i^{(L)}(\boldsymbol{x},n)\right)^2 \bigg| f_{\cdot}^{(L-1)}(\boldsymbol{x},n)\right]^2\right],$$

which, however, is not helpful, since the distribution of $f_{\cdot}^{(L-1)}(\boldsymbol{x},n)$ is not Gaussian. The only exception is the case of two hidden layers, i.e. $L = 2$, where the conditioning argument allows to bound the expectations, being the random variable $f_{\cdot}^{(L-1)}(\boldsymbol{x},n) = f_{\cdot}^{(1)}(\boldsymbol{x})$ distributed as a Gaussian distribution.

We conclude by presenting an application of Theorem 7 for $L = 2$, making more explicit the estimate of $d_{W_1}(F,N)$. For simplicity, we assume a NN without bias. Given an input $\boldsymbol{x} \in \mathbb{R}^d$, then we write

$$F = \sigma_w n^{-1/2} \sum_{i=1}^n w_i \tau\left(\sigma_w n^{-1/2} \sum_{j=1}^n w_{i,j}^{(1)} \tau(\sigma_w \langle w_j^{(0)}, \boldsymbol{x}\rangle_{\mathbb{R}^d})\right). \tag{19}$$

As before, $F \stackrel{d}{=} \tilde{F}$, where

$$\tilde{F} := \sigma_w n^{-1/2} \sum_{i=1}^n w_i \tau\left(\sigma_w n^{-1/2} \sum_{j=1}^n w_{i,j}^{(1)} \tau(\Gamma Y_j)\right),$$

with $\Gamma^2 = \sigma_w^2 \|\boldsymbol{x}\|_2^2$ and $Y_j \stackrel{d}{=} w_{j,i}^{(1)} \stackrel{d}{=} w_j \sim \mathcal{N}(0,1)$ for all $i,j \in [n]$. The next theorem applies Theorem 7.

**Theorem 8** *Let $F$ be the NN output (19) with $\tau \in C^2(\mathbb{R})$ such that $|\tau(x)| \leq \alpha + \beta|x|^\gamma$ and $\left|\frac{d^l}{dx^l}\tau(x)\right| \leq \alpha + \beta|x|^\gamma$ for $l = 1,2$ and some $\alpha,\beta,\gamma \geq 0$. If $N \sim \mathcal{N}(0,\sigma^2)$ with $\sigma^2 = \mathrm{Var}[F]$, then for any $n \geq 1$*

$$d_M(F,N) \leq c_M \frac{K_1}{\sqrt[4]{n}}, \tag{20}$$

*where $K_1$ is a constant independent of $n$ and $d$ which depends on some expectations of the standard Gaussian law and can be computed explicitly, and $c_M$ is as in Theorem 4.*

See Appendix F for the proof of Theorem 8. Theorem 8 can be adapted to a NN with $p$ inputs, along the same lines as Theorem 6 adapted Theorem 5. The next theorem is a further application of Theorem 7.

**Theorem 9** *Let $F = [F_1 \ \ldots \ F_p]$ with $F_i$ being the NN output (19), for $i = 1,\ldots,p$, with $\tau \in C^2(\mathbb{R})$ such that $|\tau(x)| \leq \alpha + \beta|x|^\gamma$ and $\left|\frac{d^l}{dx^l}\tau(x)\right| \leq \alpha + \beta|x|^\gamma$ for $l = 1,2$ and some $\alpha,\beta,\gamma \geq 0$. Furthermore, let $C$ be the covariance matrix of $F$. If $N = [N_1 \ \ldots \ N_p] \sim \mathcal{N}(0,C)$, then for any $n \geq 1$*

$$d_{W_1}(F,N) \leq 2K_p \frac{\lambda_1(C)}{\lambda_p(C)} \sqrt{\frac{p}{\sqrt{n}}} \tag{21}$$

*where $\lambda_1(C)$ and $\lambda_p(C)$ are the maximum and the minimum eigenvalues of $C$, respectively, and where $K_p$ is a constant independent of $n$ and $d$ which depends on some expectations of the standard Gaussian law and can be computed explicitly.*

See Appendix F for the proof of Theorem 9. Theorem 9 shows how a direct use of second-order Poincaré inequalities on the NN's output does not allow to achieve the rate of convergence $n^{-1/2}$ established in Basteri and Trevisan (2022), Apollonio et al. (2023) and Favaro et al. (2023), which is itself worst than the rate $n^{-1}$ obtained in Trevisan (2023). In particular, for linearly-bounded activation functions, the direct use of second-order Gaussian Poincaré leads to the rate $\mathcal{O}(\sqrt{p/\sqrt{n}})$, and such a rate can not be improved, since assuming $\tau = id$ leads to the same rate. We refer to Appendix F for details. Based on these observations, for a deep Gaussian NN of depth $L \geq 1$, we conjecture that Theorem 7 leads to the rate of convergence $\mathcal{O}(L\sqrt{p/\sqrt{n}})$, which is worse than the rate of convergence established, for instance, in the work of Basteri and Trevisan (2022), that is $\mathcal{O}(L\sqrt{p/n})$. As we proved for $\tau = id$, there are no chances to avoid this worsening in the rate of convergence when second order Poincaré inequalities are applied directly to the NN's output to establish a QCLT.

## 5. Discussion

We investigated the use of second-order Poincaré inequalities to establish QCLTs for the NN's output $f^{(L+1)}(\mathbf{X}, n)$, showing their pros and cons in such a new field of application. For shallow Gaussian NNs, i.e. $L = 1$, Theorem 4, Theorem 5 and Theorem 6 show how second-order Poincaré inequalities provide a powerful tool: they reduce the problem of establishing QCLTs to the algebraic problem of computing the gradient and the Hessian of the NN's output, which is straightforward for shallow NNs, while achieving the same rate of convergence $n^{-1/2}$ as in Basteri and Trevisan (2022) and Apollonio et al. (2023), which, however, is known to be suboptimal from Favaro et al. (2023) and Trevisan (2023). For deep Gaussian NNs. i.e. $L \geq 2$, Theorem 8 and Theorem 9 show how the use of second-order Poincaré inequalities become problematic: while they still reduce the problem of establishing QCLTs to the algebraic problem of computing the gradient and the Hessian of the NN's output, the rate of convergence worsen to $n^{-1/4}$, thus not achieving rate of convergence $n^{-1/2}$ as in Basteri and Trevisan (2022) and Apollonio et al. (2023), which is known to be suboptimal from Favaro et al. (2023) and Trevisan (2023). We interpret such a worsening in the rate of convergence as peculiar feature of the application of second-order Poincaré inequalities, this being the "cost" for having a straightforward, and general, procedure to obtain QCLTs for $f^{(L+1)}(\mathbf{X}, n)$. In general, the approaches of Basteri and Trevisan (2022), Apollonio et al. (2023), Favaro et al. (2023) and Trevisan (2023) consist of a careful analysis of the NN, by combining non-trivial probabilistic tools with ad-hoc techniques that rely on the recursive definition of the NN, typically by means of an induction argument over the layers, which makes unclear if and how they still apply to other NN's architectures. Instead, the use of second-order Poincaré inequalities rely only on the fact that the NN is a functional of a Gaussian process, thus reducing the problem of establishing QCLTs to the algebraic problem of computing gradients and Hessians, which still applies to other NN's architectures.

Regarding the applicability of our method to other neural network architectures, let us consider Theorem 2. This theorem provides an upper bound between a twice-differentiable function of independent Gaussian variables and a Gaussian random variable with the same mean and variance. In our context, the Gaussian weights of the neural network assume the role of the Gaussian random variables, while the recursive definition of the neural network represents the expression of the twice-differentiable function. We argue that using this approach to establish upper bounds is not exclusive to deep fully-connected feed-forward NNs, unlike other approaches recently proposed by Trevisan (2023), Apollonio et al. (2023), Favaro et al. (2023). The versatility of Theorem 2 allows it to be applied, in principle, to any NN's architecture where we can express the output as a twice-differentiable function of independent Gaussian random variables. However, note that this approach assumes the function $f$ to be twice differentiable, which may not hold true for certain NN's architectures of interest. We tried to relax this assumption to include differentiable or just continuous activations, like the famous ReLU function (i.e. $\mathrm{ReLU}(x) = \max\{0, x\}$) which is excluded from our analysis, but in vain. Some results in this direction can be found in Eldan et al. (2021), though using Rademacher weights for the hidden layer.

Further, because of the generality of the proposed approach, we expect it to lead to rate that are suboptimal with respect to the corresponding rates that would be obtained by developing ad-hoc approaches. However, to date, establishing QCLTs beyond fully-connected feed-forward NNs is still an open problem, though CLT are available in the literature.

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

## Appendix A. Second-order Poincaré inequality for functionals of Gaussian fields

We present a brief overview of the main results of Vidotto (2020), of which Theorem 2 and Theorem 3 are special cases for random variables in $\mathbb{R}^d$. The main results of Vidotto (2020) improve on previous results of Nourdin et al. (2009), and such an improvement is obtained by using the Mehler representation of the Ornstein–Uhlenbeck semigroup, which was exploited in Last et al. (2016) to obtain second-order Poincaré inequalities for Poisson functionals. According to the Mehler formula, if $F \in L^1$, $X'$ is an independent copy of a random variable $X$, with $X$ and $X'$ being defined on the product probability space $(\Omega \times \Omega', \mathscr{F} \otimes \mathscr{F}', \mathbb{P} \times \mathbb{P}')$, and $P_t$ is the infinitesimal generator of the Ornstein–Uhlenbeck process then

$$P_t F = E\left[ f\left( e^{-t} X + \sqrt{1 - e^{-2t}} X' \right) \mid X \right], \quad t \geq 0.$$

Before stating (Vidotto, 2020, Theorem 2.1), it is useful to introduce some notation and definitions from Gaussian analysis and Malliavin calculus. We recall that an isonormal Gaussian process $X = \{X(h) : h \in H\}$ over $H = L^2(A, \mathscr{B}(A), \mu)$, where $(A, \mathscr{B}(A))$ is a Polish space endowed with its Borel $\sigma$-field and $\mu$ is a positive, $\sigma$-finite and non-atomic measure, is a centered Gaussian family defined on $(\Omega, \mathscr{F}, \mathbb{P})$ such that $E[X(h)X(g)] = \langle g, h \rangle_H$ for every $h, g \in H$. We denote by $L^2(\Omega; H)$ the set of $H$-valued random variables $Y$ satisfying $\mathbb{E}[\|Y\|_H^2] < \infty$. Furthermore, if $\mathcal{S}$ denotes the set of random variables of the form

$$F = f\left( X(\phi_1), \ldots, X(\phi_m) \right),$$

where $f : \mathbb{R}^m \to \mathbb{R}$ is a $C^\infty$-function such that $f$ and its partial derivatives have at most polynomial growth at infinity, and $\phi_i \in H$, for $i = 1, \ldots, m$, the Malliavin derivative of $F$ is the element of $L^2(\Omega; H)$ defined by

$$DF = \sum_{i=1}^{m} \frac{\partial f}{\partial x_i} \left( X(\phi_1), \ldots, X(\phi_m) \right) \phi_i.$$

Moreover, in analogy with $DF$, the second Malliavin derivative of $F$ is the element of $L^2(\Omega; H^{\odot})$ defined by

$$D^2 F = \sum_{i,j=1}^{m} \frac{\partial^2 f}{\partial x_i \partial x_j} \left( X(\phi_1), \ldots, X(\phi_m) \right) \phi_i \phi_j,$$

where $H^{\odot 2}$ is the second symmetric tensor power of $H$, so that $H^{\odot 2} = L_s^2\left( A^2, \mathscr{B}\left( A^2 \right), \mu^2 \right)$ is the subspace of $L^2\left( A^2, \mathscr{B}\left( A^2 \right), \mu^2 \right)$ whose elements are a.e. symmetric. Let us also define the Sobolev spaces $\mathbb{D}^{\alpha, p}, p \geq 1, \alpha = 1, 2$, which are defined as the closure of $\mathcal{S}$ with respect to the norms

$$\|F\|_{\mathbb{D}^{\alpha,p}} = \left( E\left[ |F|^p \right] + E\left[ \|DF\|_H^p + E\left[ \left\|D^2 F\right\|_{H^{\otimes 2}}^p \right] \mathbb{1}_{\{\alpha=2\}} \right)^{1/p}.$$

In particular, the Sobolev space $\mathbb{D}^{\alpha, p}$ is typically referred to as the domain of $D^\alpha$ in $L^p(\Omega)$. Finally, for every $1 \leq m \leq n$, every $r = 1, \ldots, m$, every $f \in L^2\left( A^m, \mathscr{B}\left( A^m \right), \mu^m \right)$ and every

$g \in L^2\left(A^n, \mathscr{B}\left(A^n\right), \mu^n\right)$, the $r$-th contraction $f \otimes_r g : A^{n+m-2r} \to \mathbb{R}$ is defined to be the following function:

$$
\begin{aligned}
f \otimes_r g\left(y_1, \ldots, y_{n+m-2r}\right) = \int_{A^r} & f\left(x_1, \ldots, x_r, y_1, \ldots, y_{m-r}\right) \\
& \times g\left(x_1, \ldots, x_r, y_{m-r+1}, \ldots, y_{m+n-2r}\right) \mathrm{d}\mu\left(x_1\right) \cdots \mathrm{d}\mu\left(x_r\right).
\end{aligned}
$$

Now, we can state (Vidotto, 2020, Theorem 2.1), which provides a second-order Poincaré inequality for a suitable class of functionals of Gaussian fields. For random variables in $\mathbb{R}^d$, the next theorem reduces to Theorem 2.

**Theorem 10 (Vidotto (2020), Theorem 2.1)** *Let $F \in \mathbb{D}^{2,4}$ be such that $E[F] = 0$ and $E\left[F^2\right] = \sigma^2$, and let $N \sim \mathcal{N}\left(0, \sigma^2\right)$; then,*

$$
\begin{aligned}
d_M(F, N) \leq c_M \Bigg( \int_{A \times A} & \left\{ E\left[\left(\left(D^2 F \otimes_1 D^2 F\right)(x, y)\right)^2\right] \right\}^{1/2} \\
& \times \left\{ E\left[(DF(x)DF(y))^2\right] \right\}^{1/2} \mathrm{d}\mu(x)\mathrm{d}\mu(y) \Bigg)^{1/2}
\end{aligned}
$$

*where $M \in \{TV, KS, W_1\}$ and $c_{TV} = \frac{4}{\sigma^2}, c_{KS} = \frac{2}{\sigma^2}, c_{W_1} = \sqrt{\frac{8}{\sigma^2 \pi}}$.*

The novelty of Theorem 10 lies in the fact that the upper bound is directly computable, making the approach of Vidotto (2020) very appealing for concrete applications of the Gaussian approximation. In particular, Theorem 10 improves over previous results of Chatterjee (2009) and Nourdin et al. (2009). Now, we can state (Vidotto, 2020, Theorem 2.3), which provides a generalization of Theorem 10 to multidimensional functionals. For random variables in $\mathbb{R}^d$, the next theorem reduces to Theorem 3.

**Theorem 11 (Vidotto (2020), Theorem 2.3)** *Let $F = [F_1 \ldots F_p]$, where, for each $i = 1, \ldots, p, F_i \in \mathbb{D}^{2,4}$ is such that $E[F_i] = 0$ and $E[F_i F_j] = c_{ij}$, with $C = \{c_{ij}\}_{i,j=1,\ldots,p}$ a symmetric and positive definite matrix. Let $N \sim \mathcal{N}(0, C)$, then we have that $d_{W_1}(F, N) \leq 2\sqrt{p}\left\|C^{-1}\right\|_{op} \|C\|_{op} \times$*

$$
\sqrt{\sum_{i,k=1}^{p} \int_{A \times A} \left\{ E\left[\left(\left(D^2 F_i \otimes_1 D^2 F_i\right)(x, y)\right)^2\right] \right\}^{1/2} \left\{ E\left[(DF_k(x)DF_k(y))^2\right] \right\}^{1/2} \mathrm{d}\mu(x)\mathrm{d}\mu(y)}.
$$

## Appendix B. Proof of Theorem 4

To apply Theorem 2, we start by computing some first and second order partial derivatives. That is,

$$
\begin{cases}
\dfrac{\partial F}{\partial w_j} = n^{-1/2}\tau(w_j^{(0)}) \\[2mm]
\dfrac{\partial F}{\partial w_j^{(0)}} = n^{-1/2}w_j\tau'(w_j^{(0)}) \\[2mm]
\nabla^2_{w_j,w_i}F = 0 \\[2mm]
\nabla^2_{w_j,w_i^{(0)}}F = n^{-1/2}\tau'(w_j^{(0)})\delta_{ij} \\[2mm]
\nabla^2_{w_j^{(0)},w_i^{(0)}}F = n^{-1/2}w_j\tau''(w_j^{(0)})\delta_{ij}
\end{cases}
$$

with $i,j = 1\ldots n$. Then, by a direct application of Theorem 2, we obtain the following preliminary estimate

$$
d_M(F,N) \le c_M \Bigg\{ \sum_{j=1}^n 2\left\{ \mathbb{E}\left[\left(\langle \nabla^2_{w_j,\cdot}F, \nabla^2_{w_j^{(0)},\cdot}F\rangle\right)^2\right] \mathbb{E}\left[\left(\frac{\partial F}{\partial w_j}\frac{\partial F}{\partial w_j^{(0)}}\right)^2\right]\right\}^{1/2}
$$
$$
+ \left\{\mathbb{E}\left[\left(\langle \nabla^2_{w_j,\cdot}F, \nabla^2_{w_j,\cdot}F\rangle\right)^2\right]\mathbb{E}\left[\left(\frac{\partial F}{\partial w_j}\frac{\partial F}{\partial w_j}\right)^2\right]\right\}^{1/2}
$$
$$
+ \left\{\mathbb{E}\left[\left(\langle \nabla^2_{w_j^{(0)},\cdot}F, \nabla^2_{w_j^{(0)},\cdot}F\rangle\right)^2\right]\mathbb{E}\left[\left(\frac{\partial F}{\partial w_j^{(0)}}\frac{\partial F}{\partial w_j^{(0)}}\right)^2\right]\right\}^{1/2}\Bigg\}^{1/2},
$$

which can be further developed. In particular, we can write the right-hand side of the previous estimate as

$$
c_M \Bigg\{ \sum_{j=1}^n 2\left\{\mathbb{E}\left[\left(\frac{1}{n}w_j\tau'\left(w_j^{(0)}\right)\tau''\left(w_j^{(0)}\right)\right)^2\right]\mathbb{E}\left[\left(\frac{1}{n}w_j\tau\left(w_j^{(0)}\right)\tau'\left(w_j^{(0)}\right)\right)^2\right]\right\}^{1/2}
$$
$$
+ \left\{\mathbb{E}\left[\left(\frac{1}{\sqrt{n}}\tau'\left(w_j^{(0)}\right)\right)^4\right]\mathbb{E}\left[\left(\frac{1}{\sqrt{n}}\tau\left(w_j^{(0)}\right)\right)^4\right]\right\}^{1/2}
$$
$$
+ \left\{\mathbb{E}\left[\left(\frac{1}{n}\left\{\tau'\left(w_j^{(0)}\right)\right\}^2 + \frac{1}{n}w_j^2\left\{\tau''\left(w_j^{(0)}\right)\right\}^2\right)^2\right]\mathbb{E}\left[\left(\frac{1}{\sqrt{n}}w_j\tau'\left(w_j^{(0)}\right)\right)^4\right]\right\}^{1/2}\Bigg\}^{1/2}
$$
$$
\overset{(\mathbb{E}[w_j^2]=1)}{=} \frac{c_M}{n}\Bigg\{\sum_{j=1}^n 2\left\{\mathbb{E}\left[\left(\tau'\left(w_j^{(0)}\right)\tau''\left(w_j^{(0)}\right)\right)^2\right]\mathbb{E}\left[\left(\tau\left(w_j^{(0)}\right)\tau'\left(w_j^{(0)}\right)\right)^2\right]\right\}^{1/2}
$$
$$
+ \left\{\mathbb{E}\left[\left(\tau'\left(w_j^{(0)}\right)\right)^4\right]\mathbb{E}\left[\left(\tau\left(w_j^{(0)}\right)\right)^4\right]\right\}^{1/2}
$$

$$+ \left\{ \mathbb{E}\left[\left(\left\{\tau'\left(w_j^{(0)}\right)\right\}^2 + w_j^2\left\{\tau''\left(w_j^{(0)}\right)\right\}^2\right)^2\right] \mathbb{E}\left[\left(w_j\tau'\left(w_j^{(0)}\right)\right)^4\right]\right\}^{1/2} \right\}^{1/2}$$

$$\overset{(iid)}{=} \frac{c_M}{\sqrt{n}} \left\{ 2\left\{\mathbb{E}\left[(\tau'(Z)\tau''(Z))^2\right]\mathbb{E}\left[(\tau(Z)\tau'(Z))^2\right]\right\}^{1/2} + \left\{\mathbb{E}\left[(\tau'(Z))^4\right]\mathbb{E}\left[(\tau(Z))^4\right]\right\}^{1/2} \right.$$

$$\left. + \left\{\mathbb{E}\left[\left(\left\{\tau'(Z)\right\}^2 + w_j^2\left\{\tau''(Z)\right\}^2\right)^2\right]\mathbb{E}\left[\left(w_j\tau'(Z)\right)^4\right]\right\}^{1/2} \right\}^{1/2}$$

$$\overset{(iid)}{=} \frac{c_M}{\sqrt{n}} \left\{ 2\left\{\mathbb{E}\left[(\tau'(Z)\tau''(Z))^2\right]\mathbb{E}\left[(\tau(Z)\tau'(Z))^2\right]\right\}^{1/2} + \left\{\mathbb{E}\left[(\tau'(Z))^4\right]\mathbb{E}\left[(\tau(Z))^4\right]\right\}^{1/2} \right.$$

$$\left. + \left\{\mathbb{E}\left[\left(\left\{\tau'(Z)\right\}^2 + w_j^2\left\{\tau''(Z)\right\}^2\right)^2\right]\mathbb{E}\left[\left(w_j\tau'(Z)\right)^4\right]\right\}^{1/2} \right\}^{1/2}$$

$$= \frac{c_M}{\sqrt{n}} \left\{ 2\left\{\mathbb{E}\left[(\tau'(Z)\tau''(Z))^2\right]\mathbb{E}\left[(\tau(Z)\tau'(Z))^2\right]\right\}^{1/2} + \left\{\mathbb{E}\left[(\tau'(Z))^4\right]\mathbb{E}\left[(\tau(Z))^4\right]\right\}^{1/2} \right.$$

$$\left. + \left\{\left(\mathbb{E}\left[\left\{\tau'(Z)\right\}^4\right] + 2\mathbb{E}\left[\left\{\tau'(Z)\right\}^2\left\{\tau''(Z)\right\}^2\right] + 3\mathbb{E}\left[\left\{\tau''(Z)\right\}^4\right]\right)3\mathbb{E}\left[\left\{\tau'(Z)\right\}^4\right]\right\}^{1/2} \right\}^{1/2}$$

$$= \frac{c_M}{\sqrt{n}} \left\{ 2\left\{\mathbb{E}\left[|\tau'(Z)|^2|\tau''(Z)|^2\right]\mathbb{E}\left[|\tau(Z)|^2|\tau'(Z)|^2\right]\right\}^{1/2} + \left\{\mathbb{E}\left[|\tau'(Z)|^4\right]\mathbb{E}\left[|\tau(Z)|^4\right]\right\}^{1/2} \right.$$

$$\left. + \left\{\left(\mathbb{E}\left[|\tau'(Z)|^4\right] + 2\mathbb{E}\left[|\tau'(Z)|^2|\tau''(Z)|^2\right] + 3\mathbb{E}\left[|\tau''(Z)|^4\right]\right)3\mathbb{E}\left[|\tau'(Z)|^4\right]\right\}^{1/2} \right\}^{1/2},$$

where $Z \sim \mathcal{N}(0,1)$. Now, since $\tau$ is polynomially bounded and the square root is an increasing function,

$$d_M(F,N) \leq \frac{c_M}{\sqrt{n}} \left\{ 2\left\{\mathbb{E}\left[(\alpha+\beta|Z|^\gamma)^4\right]\mathbb{E}\left[(\alpha+\beta|Z|^\gamma)^4\right]\right\}^{1/2} \right.$$

$$+ \left\{\mathbb{E}\left[(\alpha+\beta|Z|^\gamma)^4\right]\mathbb{E}\left[(\alpha+\beta|Z|^\gamma)^4\right]\right\}^{1/2}$$

$$\left. + \left\{18\mathbb{E}\left[(\alpha+\beta|Z|^\gamma)^4\right]\mathbb{E}\left[(\alpha+\beta|Z|^\gamma)^4\right]\right\}^{1/2} \right\}^{1/2}$$

$$= \frac{c_M}{\sqrt{n}}\sqrt{3\sqrt{2}+3}\left\{\mathbb{E}\left[(\alpha+\beta|Z|^\gamma)^4\right]\right\}^{1/2} = \frac{c_M}{\sqrt{n}}\sqrt{3(1+\sqrt{2})}\|\alpha+\beta|Z|^\gamma\|_{L_4}^2,$$

where $Z \sim \mathcal{N}(0,1)$.

## Appendix C. Proof of Theorem 5

As stated in the main body, we will make use of the fact that

$$F \overset{d}{=} \tilde{F} := n^{-1/2}\sigma_w\sum_{j=1}^n w_j\tau\left(\Gamma \cdot Y_j\right) + \sigma_b \cdot b,$$

where $\Gamma = \sigma_w^2\|x\|^2 + \sigma_b^2$. First, it is easy to see that $\mathbb{E}[F] = 0$ and that

$$\sigma^2 = \mathrm{Var}[F] = \mathrm{Var}[\tilde{F}] = \sigma_w^2\mathbb{E}_{Z\sim\mathcal{N}(0,1)}\left[\tau^2\left(\Gamma Z\right)\right] + \sigma_b^2.$$

Then we have that $d_M(F, N) = d_M(\tilde{F}, N)$, where $N \sim \mathcal{N}(0, \sigma^2)$, hence it is enough to apply Theorem 2 to $\tilde{F}$. To this aim, we compute again the gradient and the Hessian of $\tilde{F}$, noticing that the only difference with the Shallow case lies in the presence of an extra factor $\sigma_w$ in front of the sum, an extra factor of $\Gamma$ inside the activation and the bias term $\sigma_b^2 b$:

$$
\begin{cases}
\frac{\partial \tilde{F}}{\partial b} = \sigma_b \\[2mm]
\frac{\partial \tilde{F}}{\partial w_j} = n^{-1/2} \sigma_w \cdot \tau\left(\Gamma Y_j\right) \\[2mm]
\frac{\partial \tilde{F}}{\partial Y_j} = n^{-1/2} \sigma_w \Gamma \cdot w_j \cdot \tau'\left(\Gamma Y_j\right) \\[2mm]
\nabla_{b,\cdot}^2 \tilde{F} = 0 \\[2mm]
\nabla_{w_j, w_i}^2 \tilde{F} = 0 \\[2mm]
\nabla_{w_j, Y_i}^2 \tilde{F} = n^{-1/2} \sigma_w \Gamma \cdot \tau'\left(\Gamma Y_j\right) \delta_{ij} \\[2mm]
\nabla_{Y_j, Y_i}^2 \tilde{F} = n^{-1/2} \sigma_w \Gamma^2 \cdot w_j \cdot \tau''\left(\Gamma Y_j\right) \delta_{ij}
\end{cases}
$$

It is interesting to notice that since the row of the Hessian corresponding to the bias term $b$ contains all zeros, then the bound given by Theorem 2 is exactly the same as the one at the beginning of the proof of Theorem 4, with the only difference that now the expectations depend also on $\Gamma$ and $\sigma_w$. More precisely, we have that

$$d_M(F, N) = d_M\left(\tilde{F}, N\right) \leq$$

$$\leq c_M \left\{ \sum_{j=1}^n 2 \left\{ \mathbb{E}\left[ \left( \langle \nabla_{w_j,\cdot}^2 \tilde{F}, \nabla_{Y_j,\cdot}^2 \tilde{F} \rangle \right)^2 \right] \cdot \mathbb{E}\left[ \left( \frac{\partial \tilde{F}}{\partial w_j} \cdot \frac{\partial \tilde{F}}{\partial Y_j} \right)^2 \right] \right\}^{1/2} \right.$$

$$+ \left\{ \mathbb{E}\left[ \left( \langle \nabla_{w_j,\cdot}^2 \tilde{F}, \nabla_{w_j,\cdot}^2 \tilde{F} \rangle \right)^2 \right] \cdot \mathbb{E}\left[ \left( \frac{\partial \tilde{F}}{\partial w_j} \cdot \frac{\partial \tilde{F}}{\partial w_j} \right)^2 \right] \right\}^{1/2}$$

$$\left. + \left\{ \mathbb{E}\left[ \left( \langle \nabla_{Y_j,\cdot}^2 \tilde{F}, \nabla_{Y_j,\cdot}^2 \tilde{F} \rangle \right)^2 \right] \cdot \mathbb{E}\left[ \left( \frac{\partial \tilde{F}}{\partial Y_j} \cdot \frac{\partial \tilde{F}}{\partial Y_j} \right)^2 \right] \right\}^{1/2} \right\}^{1/2}$$

$$= c_M \left\{ \sum_{j=1}^n 2 \left\{ \mathbb{E}\left[ \left( \frac{1}{n} \sigma_w^2 \Gamma^3 w_j \tau'\left(\Gamma Y_j\right) \tau''\left(\Gamma Y_j\right) \right)^2 \right] \cdot \mathbb{E}\left[ \left( \frac{1}{n} \sigma_w^2 \Gamma w_j \tau\left(\Gamma Y_j\right) \tau'\left(\Gamma Y_j\right) \right)^2 \right] \right\}^{1/2} \right.$$

$$\left. + \left\{ \mathbb{E}\left[ \left( \frac{1}{\sqrt{n}} \sigma_w \Gamma \tau'\left(\Gamma Y_j\right) \right)^4 \right] \cdot \mathbb{E}\left[ \left( \frac{1}{\sqrt{n}} \sigma_w \tau\left(\Gamma Y_j\right) \right)^4 \right] \right\}^{1/2} \right.$$

$$
+\left\{\mathbb{E}\left[\left(\frac{1}{n}\sigma_w^2\Gamma^2\left\{\tau'\left(\Gamma Y_j\right)\right\}^2+\frac{1}{n}\sigma_w^2\Gamma^4 w_j^2\left\{\tau''\left(\Gamma Y_j\right)\right\}^2\right)^2\right]\mathbb{E}\left[\left(\frac{1}{\sqrt{n}}\sigma_w\Gamma w_j\tau'\left(\Gamma Y_j\right)\right)^4\right]\right\}^{1/2}\right\}^{1/2}
$$

$$
\overset{\mathbb{E}w_j^2=1}{=}\frac{c_M}{n}\sigma_w^2\left\{\sum_{j=1}^n 2\Gamma^4\left\{\mathbb{E}\left[\left(\tau'\left(\Gamma Y_j\right)\tau''\left(\Gamma Y_j\right)\right)^2\right]\cdot\mathbb{E}\left[\left(\tau\left(\Gamma Y_j\right)\tau'\left(\Gamma Y_j\right)\right)^2\right]\right\}^{1/2}
$$

$$
+\Gamma^2\left\{\mathbb{E}\left[\left(\tau'\left(\Gamma Y_j\right)\right)^4\right]\cdot\mathbb{E}\left[\left(\tau\left(\Gamma Y_j\right)\right)^4\right]\right\}^{1/2}
$$

$$
+\left\{\mathbb{E}\left[\left(\Gamma^2\left\{\tau'\left(\Gamma Y_j\right)\right\}^2+\Gamma^4 w_j^2\left\{\tau''\left(\Gamma Y_j\right)\right\}^2\right)^2\right]\cdot\mathbb{E}\left[\left(\Gamma w_j\tau'\left(\Gamma Y_j\right)\right)^4\right]\right\}^{1/2}\right\}^{1/2}
$$

$$
\overset{iid}{=}\frac{c_M}{\sqrt{n}}\sigma_w^2\left\{2\Gamma^4\left\{\mathbb{E}\left[\left(\tau'\left(\Gamma Z\right)\tau''\left(\Gamma Z\right)\right)^2\right]\cdot\mathbb{E}\left[\left(\tau\left(\Gamma Z\right)\tau'\left(\Gamma Z\right)\right)^2\right]\right\}^{1/2}
$$

$$
+\Gamma^2\left\{\mathbb{E}\left[\left(\tau'\left(\Gamma Z\right)\right)^4\right]\cdot\mathbb{E}\left[\left(\tau\left(\Gamma Z\right)\right)^4\right]\right\}^{1/2}
$$

$$
+\left\{\mathbb{E}\left[\left(\Gamma^2\left\{\tau'\left(\Gamma Z\right)\right\}^2+\Gamma^4 w_j^2\left\{\tau''\left(\Gamma Z\right)\right\}^2\right)^2\right]\cdot\mathbb{E}\left[\left(\Gamma w_j\tau'\left(\Gamma Z\right)\right)^4\right]\right\}^{1/2}\right\}^{1/2}
$$

$$
=\frac{c_M}{\sqrt{n}}\sigma_w^2\left\{2\Gamma^4\left\{\mathbb{E}\left[\left(\tau'\left(\Gamma Z\right)\tau''\left(\Gamma Z\right)\right)^2\right]\cdot\mathbb{E}\left[\left(\tau\left(\Gamma Z\right)\tau'\left(\Gamma Z\right)\right)^2\right]\right\}^{1/2}
$$

$$
+\Gamma^2\left\{\mathbb{E}\left[\left(\tau'\left(\Gamma Z\right)\right)^4\right]\cdot\mathbb{E}\left[\left(\tau\left(\Gamma Z\right)\right)^4\right]\right\}^{1/2}
$$

$$
+\left\{\left(\Gamma^4\mathbb{E}\left[\left\{\tau'\left(\Gamma Z\right)\right\}^4\right]+2\Gamma^6\mathbb{E}\left[\left\{\tau'\left(\Gamma Z\right)\right\}^2\left\{\tau''\left(\Gamma Z\right)\right\}^2\right]+3\Gamma^8\mathbb{E}\left[\left\{\tau''\left(\Gamma Z\right)\right\}^4\right]\right)\right.
$$

$$
\left.\times 3\Gamma^4\cdot\mathbb{E}\left[\left\{\tau'\left(\Gamma Z\right)\right\}^4\right]\right\}^{1/2}\right\}^{1/2}
$$

$$
=\frac{c_M}{\sqrt{n}}\sigma_w^2\left\{2\Gamma^4\left\{\mathbb{E}\left[|\tau'\left(\Gamma Z\right)|^2|\tau''\left(\Gamma Z\right)|^2\right]\cdot\mathbb{E}\left[|\tau\left(\Gamma Z\right)|^2|\tau'\left(\Gamma Z\right)|^2\right]\right\}^{1/2}
$$

$$
+\Gamma^2\left\{\mathbb{E}\left[|\tau'\left(\Gamma Z\right)|^4\right]\cdot\mathbb{E}\left[|\tau\left(\Gamma Z\right)|^4\right]\right\}^{1/2}
$$

$$
+\left\{\left(\Gamma^4\mathbb{E}\left[|\tau'\left(\Gamma Z\right)|^4\right]+2\Gamma^6\cdot\mathbb{E}\left[|\tau'\left(\Gamma Z\right)|^2|\tau''\left(\Gamma Z\right)|^2\right]+3\Gamma^8\cdot\mathbb{E}\left[|\tau''\left(\Gamma Z\right)|^4\right]\right)\right.
$$

$$
\left.\times 3\Gamma^4\cdot\mathbb{E}\left[|\tau'\left(\Gamma Z\right)|^4\right]\right\}^{1/2}\right\}^{1/2},
$$

where $Z\sim\mathcal{N}(0,1)$. But since $\tau$ is polynomially bounded and the square root is an increasing function, we can bound this expression by

$$
\frac{c_M}{\sqrt{n}}\sigma_w^2\left\{2\Gamma^4\left\{\mathbb{E}\left[\left(\alpha+\beta|\Gamma Z|^\gamma\right)^4\right]\cdot\mathbb{E}\left[\left(\alpha+\beta|\Gamma Z|^\gamma\right)^4\right]\right\}^{1/2}
$$

$$
+\Gamma^2\left\{\mathbb{E}\left[\left(\alpha+\beta|\Gamma Z|^\gamma\right)^4\right]\cdot\mathbb{E}\left[\left(\alpha+\beta|\Gamma Z|^\gamma\right)^4\right]\right\}^{1/2}
$$

$$+\Gamma^4\left\{\sqrt{3(1+2\Gamma^2+3\Gamma^4)}\cdot\mathbb{E}\left[(\alpha+\beta|\Gamma Z|^\gamma)^4\right]\cdot\mathbb{E}\left[(\alpha+\beta|\Gamma Z|^\gamma)^4\right]\right\}^{1/2}\right\}^{1/2}$$

$$=\frac{c_M}{\sqrt{n}}\sigma_w^2\sqrt{\Gamma^2+\Gamma^4(2+\sqrt{3(1+2\Gamma^2+3\Gamma^4)})}\left\{\mathbb{E}\left[(\alpha+\beta|\Gamma Z|^\gamma)^4\right]\right\}^{1/2}$$

$$=\frac{c_M}{\sqrt{n}}\sigma_w^2\sqrt{\Gamma^2+\Gamma^4(2+\sqrt{3(1+2\Gamma^2+3\Gamma^4)})}\cdot\|\alpha+\beta|\Gamma Z|^\gamma\|_{L^4}^2,$$

where $Z\sim\mathcal{N}(0,1)$.

## Appendix D. Proof of Theorem 6

The proof is based on Theorem 3. Recall that

$$F_i:=\frac{1}{n^{1/2}}\sigma_w\sum_{j=1}^n w_j\tau(\sigma_w\langle w_j^{(0)},\boldsymbol{x_i}\rangle+\sigma_b b_j^{(0)})+\sigma_b b.$$

Since $F_1,\ldots,F_p$ are functions of the iid standard normal random variables $\{w_j,w_{jl}^{(0)},b_j^{(0)},b:$ $j=1,\ldots,n,l=1,\ldots,d\}$, then we can apply Theorem 3 to the random vector $F=[F_1\ \cdots\ F_p]$. The upper bound in (10) depends on the first and second derivatives of the $F_i$'s with respect to all their arguments. However, the derivatives with respect to $b$ give no contributions, since, for every $i=1,\ldots,p$, $\nabla_{b,.}^2 F_i$ is the zero vector. Moreover, the terms $w_j\tau(\sigma_w\langle w_j^{(0)},\boldsymbol{x_i}\rangle+\sigma_b b_j^{(0)})$ are iid, across $j$, and give the same contribution to the upper bound. Hence, we can write that

$$d_{W_1}(F,N)\le 2\sigma_w^2\sqrt{\frac{p}{n}}\|C^{-1}\|_2\|C\|_2\sqrt{\sum_{i,k=1}^p D_{ik}},$$

where

$$D_{ik}=\sum_{l,m}\left\{\mathbb{E}\left[\left(\langle\nabla_{l,.}^2\tilde{F}_i,\nabla_{m,.}^2\tilde{F}_i\rangle\right)^2\right]\right\}^{1/2}\left\{\mathbb{E}\left[\left(\nabla_l\tilde{F}_k\nabla_m\tilde{F}_k\right)^2\right]\right\}^{1/2},$$

where

$$[\tilde{F}_1\ \ldots\ \tilde{F}_p]\overset{d}{=}[w_j\tau(\sigma_w\langle w_j^{(0)},\boldsymbol{x_1}\rangle+\sigma_b b_j^{(0)})\ \ldots\ w_j\tau(\sigma_w\langle w_j^{(0)},\boldsymbol{x_p}\rangle+\sigma_b b_j^{(0)})],$$

and $\nabla_l,\nabla_m,\nabla_{l,.}^2$ and $\nabla_{m,.}^2$ denote the derivatives with respect to all the arguments. We can represent $\tilde{F}_i$ as

$$\tilde{F}_i=w\cdot\tau(Y_i),$$

where $Y_i:=\langle\tilde{w}^{(0)},\tilde{\boldsymbol{x}_i}\rangle=\sum_{s=1}^d\tilde{w}_s^{(0)}\tilde{x}_{is}$, with $\tilde{\boldsymbol{x}_i}:=[\sigma_w\boldsymbol{x}_i^T,\sigma_b]^T$, $\tilde{w}^{(0)}:=[w^{(0)T},b^{(0)}]^T$, and $w,\tilde{w}_1^{(0)},\ldots,\tilde{w}_d^{(0)},b^{(0)}$ iid standard normal random variables. The gradient and the Hessian

of $\tilde{F}$ with respect to the parameters $w$ and $\tilde{w}_s^{(0)}$ are

$$
\begin{cases}
\frac{\partial \tilde{F}_i}{\partial w} = \tau(Y_i) \\[2mm]
\frac{\partial \tilde{F}_i}{\partial w_s^{(0)}} = w\tau'(Y_i)\tilde{x}_{is} \\[2mm]
\nabla^2_{w,w} \tilde{F}_i = 0 \\[2mm]
\nabla^2_{w,\tilde{w}_s^{(0)}} \tilde{F}_i = \tau'(Y_i)\tilde{x}_{is} \\[2mm]
\nabla^2_{\tilde{w}_s^{(0)},\tilde{w}_t^{(0)}} \tilde{F}_i = w\tau''(Y_i)\tilde{x}_{is}\tilde{x}_{it}.
\end{cases}
$$

This implies that

$$
\begin{aligned}
D_{ik} &= \left\{ \mathbb{E}\left[ \left( \sum_{s=1}^{d} \nabla^2_{w,\tilde{w}_s^{(0)}} \tilde{F}_i \cdot \nabla^2_{w,\tilde{w}_s^{(0)}} \tilde{F}_i \right)^2 \right] \right\}^{1/2} \left\{ \mathbb{E}\left[ \left( \frac{\partial \tilde{F}_k}{\partial w} \cdot \frac{\partial \tilde{F}_k}{\partial w} \right)^2 \right] \right\}^{1/2} \\
&\quad + \sum_{j,j'=1}^{d} \left\{ \mathbb{E}\left[ \left( \nabla^2_{w,\tilde{w}_j^{(0)}} \tilde{F}_i \cdot \nabla^2_{w,\tilde{w}_{j'}^{(0)}} \tilde{F}_i + \sum_{s=1}^{d} \nabla^2_{\tilde{w}_j^{(0)},\tilde{w}_s^{(0)}} \tilde{F}_i \cdot \nabla^2_{\tilde{w}_{j'}^{(0)},\tilde{w}_s^{(0)}} \tilde{F}_i \right)^2 \right] \right\}^{1/2} \\
&\qquad \times \left\{ \mathbb{E}\left[ \left( \frac{\partial \tilde{F}_k}{\partial \tilde{w}_j^{(0)}} \cdot \frac{\partial \tilde{F}_k}{\partial \tilde{w}_{j'}^{(0)}} \right)^2 \right] \right\}^{1/2} \\
&\quad + 2\sum_{j=1}^{d} \left\{ \mathbb{E}\left[ \left( \sum_{s=1}^{d} \nabla^2_{w,\tilde{w}_s^{(0)}} \tilde{F}_i \cdot \nabla^2_{\tilde{w}_j^{(0)},\tilde{w}_s^{(0)}} \tilde{F}_i \right)^2 \right] \right\}^{1/2} \left\{ \mathbb{E}\left[ \left( \frac{\partial \tilde{F}_k}{\partial w} \cdot \frac{\partial \tilde{F}_k}{\partial \tilde{w}_j^{(0)}} \right)^2 \right] \right\}^{1/2} \\
&= \left\{ \mathbb{E}\left[ \left( \sum_{s=1}^{d} \tau'(Y_i)^2 \tilde{x}_{is}^2 \right)^2 \right] \right\}^{1/2} \left\{ \mathbb{E}\left[ (\tau(Y_k))^4 \right] \right\}^{1/2} \\
&\quad + \sum_{j,j'=1}^{d} \left\{ \mathbb{E}\left[ \left( \tau'(Y_i)^2 \tilde{x}_{ij}\tilde{x}_{ij'} + \sum_{s=1}^{d} w^2 \tau''(Y_i)^2 \tilde{x}_{ij}\tilde{x}_{ij'}\tilde{x}_{is}^2 \right)^2 \right] \right\}^{1/2} \left\{ \mathbb{E}\left[ (w^2\tau'(Y_k)^2 \tilde{x}_{kj}\tilde{x}_{kj'})^2 \right] \right\}^{1/2} \\
&\quad + 2\sum_{j=1}^{d} \left\{ \mathbb{E}\left[ \left( \sum_{s=1}^{d} \tau'(Y_i)\tilde{x}_{is} w\tau''(Y_i)\tilde{x}_{ij}\tilde{x}_{is} \right)^2 \right] \right\}^{1/2} \left\{ \mathbb{E}\left[ (\tau(Y_k)w\tau'(Y_k)\tilde{x}_{kj})^2 \right] \right\}^{1/2} \\
&= ||\tilde{x}_i||^2 \left\| \tau'(Y_i) \right\|_{L_4}^2 \left\| \tau(Y_k) \right\|_{L_4}^2 \\
&\quad + \sum_{j,j'=1}^{d} |\tilde{x}_{ij}\tilde{x}_{ij'}| \left\{ \mathbb{E}\left[ \left( \tau'(Y_i)^2 + w^2\tau''(Y_i)^2||\tilde{x}_i||^2 \right)^2 \right] \right\}^{1/2} \sqrt{3}|\tilde{x}_{kj}\tilde{x}_{kj'}| \left\| \tau'(Y_k) \right\|_{L^4}^2 \\
&\quad + 2\sum_{j=1}^{d} |\tilde{x}_{ij}||\tilde{x}_{kj}|||\tilde{x}_i||^2 \left\{ \mathbb{E}\left[ \left( \tau'(Y_i)\tau''(Y_i) \right)^2 \right] \right\}^{1/2} \left\{ \mathbb{E}\left[ \left( \tau(Y_k)\tau'(Y_k) \right)^2 \right] \right\}^{1/2}
\end{aligned}
$$

$$= ||\tilde{\boldsymbol{x}}_i||^2 \big\|\tau'(Y_i)\big\|_{L_4}^2 \|\tau(Y_k)\|_{L_4}^2 + \sum_{j,j'=1}^{d} \sqrt{3}|\tilde{x}_{kj}\tilde{x}_{kj'}||\tilde{x}_{ij}\tilde{x}_{ij'}|\big\|\tau'(Y_k)\big\|_{L^4}^2$$

$$\times \left\{ \big\|\tau'(Y_i)\big\|_{L_4}^4 + 3\|\tilde{\boldsymbol{x}_i}\|^4 \big\|\tau''(Y_i)\big\|_{L_4}^4 + 2\|\tilde{\boldsymbol{x}_i}\|^2 \big\|\tau'(Y_i)\tau''(Y_i)\big\|_{L_2}^2 \right\}^{1/2}$$

$$+ 2\sum_{j=1}^{d} |\tilde{x}_{ij}||\tilde{x}_{kj}|\|\tilde{\boldsymbol{x}_i}\|^2 \big\|\tau'(Y_i)\tau''(Y_i)\big\|_{L_2} \big\|\tau(Y_k)\tau'(Y_k)\big\|_{L_2}$$

$$= ||\tilde{\boldsymbol{x}}_i||^2 \big\|\tau'(Y_i)\big\|_{L_4}^2 \|\tau(Y_k)\|_{L_4}^2 + \sqrt{3}\big\|\tau'(Y_k)\big\|_{L^4}^2 \left( \sum_{j=1}^{d} |\tilde{x}_{ij}\tilde{x}_{kj}| \right)^2$$

$$\times \left\{ \big\|\tau'(Y_i)\big\|_{L_4}^4 + 3\|\tilde{\boldsymbol{x}_i}\|^4 \big\|\tau''(Y_i)\big\|_{L_4}^4 + 2\|\tilde{\boldsymbol{x}_i}\|^2 \big\|\tau'(Y_i)\tau''(Y_i)\big\|_{L_2}^2 \right\}^{1/2}$$

$$+ 2\|\tilde{\boldsymbol{x}_i}\|^2 \big\|\tau'(Y_i)\tau''(Y_i)\big\|_{L_2} \big\|\tau(Y_k)\tau'(Y_k)\big\|_{L_2} \left( \sum_{j=1}^{d} |\tilde{x}_{ij}||\tilde{x}_{kj}| \right)$$

$$\overset{\text{Holder ineq.}}{\leq} ||\tilde{\boldsymbol{x}}_i||^2 \big\|\tau'(Y_i)\big\|_{L_4}^2 \|\tau(Y_k)\|_{L_4}^2 + \sqrt{3}\big\|\tau'(Y_k)\big\|_{L^4}^2 \left( \sum_{j=1}^{d} |\tilde{x}_{ij}\tilde{x}_{kj}| \right)^2$$

$$\times \left\{ \big\|\tau'(Y_i)\big\|_{L_4}^4 + 3\|\tilde{\boldsymbol{x}_i}\|^4 \big\|\tau''(Y_i)\big\|_{L_4}^4 + 2\|\tilde{\boldsymbol{x}_i}\|^2 \big\|\tau'(Y_i)\big\|_{L_4}^2 \big\|\tau''(Y_i)\big\|_{L_4}^2 \right\}^{1/2}$$

$$+ 2\|\tilde{\boldsymbol{x}_i}\|^2 \big\|\tau'(Y_i)\big\|_{L_4} \big\|\tau''(Y_i)\big\|_{L_4} \|\tau(Y_k)\|_{L_4} \big\|\tau'(Y_k)\big\|_{L_4} \left( \sum_{j=1}^{d} |\tilde{x}_{ij}||\tilde{x}_{kj}| \right)$$

$$\overset{\text{polynom. bounded}}{\leq} ||\tilde{\boldsymbol{x}}_i||^2 \|\alpha + \beta|Y_i|^\gamma\|_{L_4}^2 \|\alpha + \beta|Y_k|^\gamma\|_{L_4}^2$$

$$+ \sqrt{3}\left\{ (1 + 2\|\tilde{\boldsymbol{x}_i}\|^2 + 3\|\tilde{\boldsymbol{x}_i}\|^4)\|\alpha + \beta|Y_i|^\gamma\|_{L_4}^4 \right\}^{1/2} \|\alpha + \beta|Y_k|^\gamma\|_{L^4}^2 \left( \sum_{j=1}^{d} |\tilde{x}_{ij}\tilde{x}_{kj}| \right)^2$$

$$+ 2\|\tilde{\boldsymbol{x}_i}\|^2 \|\alpha + \beta|Y_i|^\gamma\|_{L_4}^2 \|\alpha + \beta|Y_k|^\gamma\|_{L_4}^2 \left( \sum_{j=1}^{d} |\tilde{x}_{ij}\tilde{x}_{kj}| \right)$$

$$= \left\{ ||\tilde{\boldsymbol{x}}_i||^2 + \sqrt{3(1 + 2\|\tilde{\boldsymbol{x}_i}\|^2 + 3\|\tilde{\boldsymbol{x}_i}\|^4)} \left( \sum_{j=1}^{d} |\tilde{x}_{ij}\tilde{x}_{kj}| \right)^2 + 2\|\tilde{\boldsymbol{x}_i}\|^2 \left( \sum_{j=1}^{d} |\tilde{x}_{ij}\tilde{x}_{kj}| \right) \right\}$$

$$\times \|\alpha + \beta|Y_i|^\gamma\|_{L_4}^2 \|\alpha + \beta|Y_k|^\gamma\|_{L_4}^2.$$

Now, traducing everything back to the original variables $\{\boldsymbol{x}_i\}_{i \in [d]}$, we have that

$$\begin{cases} \sum_{j=1}^d |\tilde{x}_{ij}||\tilde{x}_{kj}| = \sigma_w^2 \sum_{j=1}^d |x_{ij}||x_{kj}| + \sigma_b^2 =: \Gamma_{ik} \\ ||\tilde{\boldsymbol{x}}_i||^2 = \sigma_w^2 ||\boldsymbol{x}_i||^2 + \sigma_b^2 =: \Gamma_i^2. \end{cases}$$

Hence,

$$D_{ik} \leq (\Gamma_i^2 + \sqrt{3(1 + 2\Gamma_i^2 + 3\Gamma_i^4)}\Gamma_{ik}^2 + 2\Gamma_i^2\Gamma_{ik})\|\alpha + \beta|Y_i|^\gamma\|_{L_4}^2\|\alpha + \beta|Y_k|^\gamma\|_{L_4}^2,$$

with $Y \sim \mathcal{N}(0, \sigma_b^2 \mathbf{X}^T \mathbf{X} + \sigma_b^2 \mathbf{1}\mathbf{1}^T)$. Summing over all possible $i, k = 1, \ldots, p$ and taking the square root leads to

$$d_{W_1}(F, N) \leq 2\sigma_w^2 \frac{\lambda_1(C)}{\lambda_p(C)} \sqrt{\frac{p}{n}} \tilde{K},$$

with

$$\tilde{K} = \left\{ \sum_{i,k=1}^p (\Gamma_i^2 + \sqrt{3(1 + 2\Gamma_i^2 + 3\Gamma_i^4)}\Gamma_{ik}^2 + 2\Gamma_i^2\Gamma_{ik})\|\alpha + \beta|Y_i|^\gamma\|_{L^4}^2\|\alpha + \beta|Y_k|^\gamma\|_{L^4}^2 \right\}^{1/2}$$

$$= \left\{ \sum_{i,k=1}^p (\Gamma_i^2 + \sqrt{3(1 + 2\Gamma_i^2 + 3\Gamma_i^4)}\Gamma_{ik}^2 + 2\Gamma_i^2\Gamma_{ik})\|\alpha + \beta|\Gamma_i Z|^\gamma\|_{L^4}^2\|\alpha + \beta|\Gamma_k Z|^\gamma\|_{L^4}^2 \right\}^{1/2},$$

with $Z \sim \mathcal{N}(0, 1)$, which concludes the proof.

## Appendix E. Gradient and Hessian for the output of a deep NN

The first step of the proofs of Theorem 8 and 9 is computing the gradient and the Hessian of $\tilde{F}$.

### E.1. $L = 2$

If $L = 2$, then

$$\tilde{F} = \sigma_w n^{-1/2} \sum_{i=1}^n w_i \tau(f_i^{(2)}(\boldsymbol{x}, n)),$$

where

$$f_i^{(2)}(\boldsymbol{x}, n) = \sigma_w n^{-1/2} \sum_{j=1}^n w_{i,j}^{(1)} \tau(f_j^{(1)}(\boldsymbol{x})),$$

$$f_j^{(1)}(\boldsymbol{x}) = \Gamma Y_j,$$

with $\Gamma^2 = \sigma_w^2 \|\boldsymbol{x}\|_2^2$. The partial derivatives are given by

$$
\begin{cases}
\dfrac{\partial \tilde{F}}{\partial w_i} = \sigma_w n^{-1/2} \tau\left(f_i^{(2)}(\boldsymbol{x}, n)\right) \\[2ex]
\dfrac{\partial \tilde{F}}{\partial w_{i,j}^{(1)}} = \left(\sigma_w n^{-1/2}\right)^2 w_i \tau\left(f_j^{(1)}(\boldsymbol{x})\right) \tau'\left(f_i^{(2)}(\boldsymbol{x}, n)\right) \\[2ex]
\dfrac{\partial \tilde{F}}{\partial Y_j} = \Gamma\left(\sigma_w n^{-1/2}\right)^2 \tau'\left(f_j^{(1)}(\boldsymbol{x})\right) \sum_{a=1}^{n} w_a w_{a,j}^{(1)} \tau'\left(f_a^{(2)}(\boldsymbol{x}, n)\right)
\end{cases}
$$

$$
\begin{cases}
\nabla^2_{w_i, w_j} \tilde{F} = 0 \\[2ex]
\nabla^2_{w_i, w_{k,j}^{(1)}} \tilde{F} = \delta_{ik} \left(\sigma_w n^{-1/2}\right)^2 \tau\left(f_j^{(1)}(\boldsymbol{x})\right) \tau'\left(f_i^{(2)}(\boldsymbol{x}, n)\right) \\[2ex]
\nabla^2_{w_i, Y_j} \tilde{F} = \Gamma\left(\sigma_w n^{-1/2}\right)^2 \tau'\left(f_j^{(1)}(\boldsymbol{x})\right) \tau'\left(f_i^{(2)}(\boldsymbol{x}, n)\right) \\[2ex]
\nabla^2_{w_{i,j}^{(1)}, w_{k,h}^{(1)}} \tilde{F} = \delta_{ik} \left(\sigma_w n^{-1/2}\right)^3 w_i \tau\left(f_j^{(1)}(\boldsymbol{x})\right) \tau\left(f_h^{(1)}(\boldsymbol{x})\right) \tau''\left(f_i^{(2)}(\boldsymbol{x}, n)\right) \\[2ex]
\nabla^2_{w_{i,j}^{(1)}, Y_k} \tilde{F} = \Gamma\left(\sigma_w n^{-1/2}\right)^2 w_i \tau'\left(f_k^{(1)}(\boldsymbol{x})\right) \left[\Gamma \sigma_w n^{-1/2} w_{i,k}^{(1)} \tau\left(f_j^{(1)}(\boldsymbol{x})\right) \tau''\left(f_i^{(2)}(\boldsymbol{x}, n)\right) \right. \\
\qquad\qquad\qquad\qquad + \left. \delta_{jk} \tau'\left(f_i^{(2)}(\boldsymbol{x}, n)\right)\right] \\[2ex]
\nabla^2_{Y_j, Y_k} \tilde{F} = \left(\Gamma \sigma_w n^{-1/2}\right)^2 \left[\sigma_w n^{-1/2} \tau'\left(f_j^{(1)}(\boldsymbol{x})\right) \tau'\left(f_k^{(1)}(\boldsymbol{x})\right) \sum_{a=1}^{n} w_a w_{a,j}^{(1)} w_{a,k}^{(1)} \tau''\left(f_a^{(2)}(\boldsymbol{x}, n)\right) \right. \\
\qquad\qquad\qquad + \delta_{jk} \tau''\left(f_j^{(1)}(\boldsymbol{x})\right) \sum_{a=1}^{n} w_a w_{a,j}^{(1)} \tau'\left(f_a^{(2)}(\boldsymbol{x}, n)\right)\Big],
\end{cases}
$$

and this for all $i, j, k \in [n]$.

### E.2. General $L$

In this section will compute the gradient and the hessian of the NN defined in (1) for a general $L$, not necessarily $L = 2$ as in the previous one. Application of Theorem 7 requires computing the gradient and the hessian of $F_i = f^{(L+1)}(\boldsymbol{x}_i)$, and it will be sufficient to use all the computations of this section with $F_i$ in place of $F$, and $\boldsymbol{x}_i$ in place of $\boldsymbol{x}$. To simplify the notation we write $f_i^{(l)}(\boldsymbol{x}) := f_i^{(l)}(\boldsymbol{x}, n)$ for every $i$ and $l$.

It is useful to start by computing the following derivatives

$$
\frac{\partial F}{\partial f_{i_L}^{(L)}(\boldsymbol{x})} = \frac{\sigma_w}{\sqrt{n}} w_{i_L} \tau'\left(f_{i_L}^{(L)}(\boldsymbol{x})\right)
$$

$$\frac{\partial f_{i_{l+1}}^{(l+1)}(\boldsymbol{x})}{\partial f_{i_l}^{(l)}(\boldsymbol{x})} = \frac{\sigma_w}{\sqrt{n}} w_{i_{l+1},i_l}^{(l)} \tau' \left( f_{i_l}^{(l)}(\boldsymbol{x}) \right) \quad \forall l \in \{1,\ldots,L-1\}$$

$$\frac{\partial f_{i_{l+1}}^{(l+1)}(\boldsymbol{x})}{\partial w_{i_l,j_l}^{(l)}} = \delta_{i_{l+1}i_l} \frac{\sigma_w}{\sqrt{n}} \tau \left( f_{j_l}^{(l)}(\boldsymbol{x}) \right) \quad \forall l \in \{1,\ldots,L-1\}$$

$$\frac{\partial f_{i_1}^{(1)}(\boldsymbol{x})}{\partial w_{i_0,j_0}^{(0)}} = \delta_{i_1 i_0} \sigma_w x_{j_0},$$

which hold true for all $i_L,\ldots,i_0,j_L,\ldots,j_1 = 1,\ldots,n$ and $j_0 = 1,\ldots,d$.
Using the chain rule, it is easy but a little tedious to compute

$$\frac{\partial F}{\partial w_{i_L}} = \frac{\sigma_w}{\sqrt{n}} \tau \left( f_{i_L}^{(L)}(\boldsymbol{x}) \right)$$

$$\frac{\partial F}{\partial w_{i_{L-1},j_{L-1}}^{(L-1)}} = \left( \frac{\sigma_w}{\sqrt{n}} \right)^2 w_{i_{L-1}} \tau' \left( f_{i_{L-1}}^{(L)}(\boldsymbol{x}) \right) \tau \left( f_{j_{L-1}}^{(L-1)}(\boldsymbol{x}) \right)$$

$$\frac{\partial F}{\partial w_{i_{L-2},j_{L-2}}^{(L-2)}} = \left( \frac{\sigma_w}{\sqrt{n}} \right)^3 \tau' \left( f_{i_{L-2}}^{(L-1)}(\boldsymbol{x}) \right) \tau \left( f_{j_{L-2}}^{(L-2)}(\boldsymbol{x}) \right) \sum_{i_L=1}^{n} w_{i_L} \tau' \left( f_{i_L}^{(L)}(\boldsymbol{x}) \right) w_{i_L,i_{L-2}}^{(L-1)}$$

$$\frac{\partial F}{\partial w_{i_l,j_l}^{(l)}} = \left( \frac{\sigma_w}{\sqrt{n}} \right)^{L-l+1} \tau' \left( f_{i_l}^{(l+1)}(\boldsymbol{x}) \right) \tau \left( f_{j_l}^{(l)}(\boldsymbol{x}) \right) \times$$

$$\times \sum_{i_L,\ldots,i_{l+2}=1}^{n} w_{i_L} \tau' \left( f_{i_L}^{(L)}(\boldsymbol{x}) \right) \left( \prod_{s=l+2}^{L-1} w_{i_{s+1},i_s}^{(s)} \tau' \left( f_{i_s}^{(s)}(\boldsymbol{x}) \right) \right) w_{i_{l+2},i_l}^{(l+1)}$$

$$\frac{\partial F}{\partial w_{i_0,j_0}^{(0)}} = \sigma_w \left( \frac{\sigma_w}{\sqrt{n}} \right)^L \tau' \left( f_{i_0}^{(1)}(\boldsymbol{x}) \right) x_{j_0} \times$$

$$\times \sum_{i_L,\ldots,i_2=1}^{n} w_{i_L} \tau' \left( f_{i_L}^{(L)}(\boldsymbol{x}) \right) \left( \prod_{s=2}^{L-1} w_{i_{s+1},i_s}^{(s)} \tau' \left( f_{i_s}^{(s)}(\boldsymbol{x}) \right) \right) w_{i_2,i_0}^{(1)}$$

for all $i_L,\ldots,i_0,j_L,\ldots,j_1 = 1,\ldots,n$, $j_0 = 1,\ldots,d$ and $l = 1,\ldots,L-3$.

As for the Hessian, we have

$$\nabla^2_{w_{i_L},w_{j_L}} F = 0$$

$$\nabla^2_{w_{i_L},w_{i_{L-1},j_{L-1}}^{(L-1)}} F = \delta_{i_L i_{L-1}} \left( \frac{\sigma_w}{\sqrt{n}} \right)^2 \tau' \left( f_{i_L}^{(L)}(\boldsymbol{x}) \right) \tau \left( f_{j_{L-1}}^{(L-1)}(\boldsymbol{x}) \right)$$

$$\nabla^2_{w_{i_L},w_{i_{L-2},j_{L-2}}^{(L-2)}} F = \left( \frac{\sigma_w}{\sqrt{n}} \right)^3 \tau' \left( f_{i_L}^{(L)}(\boldsymbol{x}) \right) \tau' \left( f_{i_{L-2}}^{(L-1)}(\boldsymbol{x}) \right) \tau \left( f_{j_{L-2}}^{(L-2)}(\boldsymbol{x}) \right) w_{i_L,i_{L-2}}^{(L-1)}$$

$$\nabla^2_{w_{i_L},w_{i_l,j_l}^{(l)}} F = \left( \frac{\sigma_w}{\sqrt{n}} \right)^{L-l+1} \tau' \left( f_{i_L}^{(L)}(\boldsymbol{x}) \right) \tau' \left( f_{i_l}^{(l+1)}(\boldsymbol{x}) \right) \tau \left( f_{j_l}^{(l)}(\boldsymbol{x}) \right) \times$$

$$\times \sum_{i_{L-1},\ldots,i_{l+2}=1}^{n} \left( \prod_{s=l+2}^{L-1} w_{i_{s+1},i_s}^{(s)} \tau' \left( f_{i_s}^{(s)}(\boldsymbol{x}) \right) \right) w_{i_{l+2},i_l}^{(l+1)}$$

$$\nabla^2_{w_{i_L}, w_{i_0,j_0}^{(0)}} F = \sigma_w \left(\frac{\sigma_w}{\sqrt{n}}\right)^L \tau'\left(f_{i_L}^{(L)}(\boldsymbol{x})\right) \tau'\left(f_{i_0}^{(1)}(\boldsymbol{x})\right) x_{j_0} \times$$

$$\times \sum_{i_{L-1},\ldots,i_2=1}^n \left(\prod_{s=2}^{L-1} w_{i_{s+1},i_s}^{(s)} \tau'\left(f_{i_s}^{(s)}(\boldsymbol{x})\right)\right) w_{i_2,i_0}^{(1)}$$

As for two generic weights $w_{i_l,j_l}^{(l)}, w_{j_m,\tilde{j}_m}^{(m)}$ for $l \in \{0,\ldots,L-1\}$, we have

$$\nabla^2_{w_{i_l,j_l}^{(l)}, w_{j_m,\tilde{j}_m}^{(m)}} F = \left(\frac{\sigma_w}{\sqrt{n}}\right)^{L-l+1} \frac{\partial}{\partial w_{j_m,\tilde{j}_m}^{(m)}} \left[\tau'\left(f_{i_l}^{(l+1)}(\boldsymbol{x})\right)\right] \tau\left(f_{j_l}^{(l)}(\boldsymbol{x})\right) \times$$

$$\times \sum_{i_L,\ldots,i_{l+2}=1}^n w_{i_L} \tau'\left(f_{i_L}^{(L)}(\boldsymbol{x})\right) \left(\prod_{s=l+2}^{L-1} w_{i_{s+1},i_s}^{(s)} \tau'\left(f_{i_s}^{(s)}(\boldsymbol{x})\right)\right) w_{i_{l+2},i_l}^{(l+1)} +$$

$$+ \left(\frac{\sigma_w}{\sqrt{n}}\right)^{L-l+1} \tau'\left(f_{i_l}^{(l+1)}(\boldsymbol{x})\right) \frac{\partial}{\partial w_{j_m,\tilde{j}_m}^{(m)}} \left[\tau\left(f_{j_l}^{(l)}(\boldsymbol{x})\right)\right] \times$$

$$\times \sum_{i_L,\ldots,i_{l+2}=1}^n w_{i_L} \tau'\left(f_{i_L}^{(L)}(\boldsymbol{x})\right) \left(\prod_{s=l+2}^{L-1} w_{i_{s+1},i_s}^{(s)} \tau'\left(f_{i_s}^{(s)}(\boldsymbol{x})\right)\right) w_{i_{l+2},i_l}^{(l+1)} +$$

$$+ \left(\frac{\sigma_w}{\sqrt{n}}\right)^{L-l+1} \tau'\left(f_{i_l}^{(l+1)}(\boldsymbol{x})\right) \tau\left(f_{j_l}^{(l)}(\boldsymbol{x})\right) \times$$

$$\times \left[\sum_{i_L,\ldots,i_{l+2}=1}^n \beta_{i_L,\ldots,i_{l+2}} \frac{\partial}{\partial w_{j_m,\tilde{j}_m}^{(m)}} \alpha_{i_L,\ldots,i_{l+2}} + \beta_{i_L,\ldots,i_{l+2}} \frac{\partial}{\partial w_{j_m,\tilde{j}_m}^{(m)}} \alpha_{i_L,\ldots,i_{l+2}}\right]$$

where

$$\alpha_{i_L,\ldots,i_{l+2}} := \left(w_{i_L} w_{i_{l+2},i_l}^{(l+1)} \prod_{s=l+2}^{L-1} w_{i_{s+1},i_s}^{(s)}\right)$$

and

$$\beta_{i_L,\ldots,i_{l+2}} := \left(\tau'\left(f_{i_L}^{(L)}(\boldsymbol{x})\right) \prod_{s=l+2}^{L-1} \tau'\left(f_{i_s}^{(s)}(\boldsymbol{x})\right)\right),$$

so that

$$\frac{\partial}{\partial w_{j_m,\tilde{j}_m}^{(m)}} \alpha_{i_L,\ldots,i_{l+2}} = \delta_{j_m,i_{m+1}} \delta_{\tilde{j}_m,i_m} \mathbb{1}\{m > l+1\} \left(\frac{w_{i_L} w_{i_{l+2},i_l}^{(l+1)}}{w_{j_m,\tilde{j}_m}^{(m)}} \prod_{s=l+2}^{L-1} w_{i_{s+1},i_s}^{(s)}\right)$$

$$+ \delta_{j_m,i_{l+2}} \delta_{\tilde{j}_m,i_l} \mathbb{1}\{m = l+1\} \left(w_{i_L} \prod_{s=l+2}^{L-1} w_{i_{s+1},i_s}^{(s)}\right),$$

$$\frac{\partial}{\partial w_{j_m,\tilde{j}_m}^{(m)}} \beta_{i_L,\ldots,i_{l+2}} = \beta_{i_L,\ldots,i_{l+2}} \sum_{s=l+2}^L \frac{1}{\tau'\left(f_{i_s}^{(s)}(\boldsymbol{x})\right)} \frac{\partial}{\partial w_{j_m,\tilde{j}_m}^{(m)}} \tau'\left(f_{i_s}^{(s)}(\boldsymbol{x})\right),$$

with

$$\frac{\partial}{\partial w_{i_m,j_m}^{(m)}}\left[\tau'\left(f_{i_l}^{(l+1)}(\boldsymbol{x})\right)\right]$$

$$=\begin{cases}
0 & \text{if } m \geq l+1 \\[2mm]
\dfrac{\sigma_w}{\sqrt{n}}\delta_{i_l i_m}\tau''\left(f_{i_l}^{(l+1)}(\boldsymbol{x})\right)\tau\left(f_{j_m}^{(m)}(\boldsymbol{x})\right) & \text{if } m = l \\[3mm]
\left(\dfrac{\sigma_w}{\sqrt{n}}\right)^2\tau''\left(f_{i_l}^{(l+1)}(\boldsymbol{x})\right)\tau'\left(f_{i_m}^{(m+1)}(\boldsymbol{x})\right)\tau\left(f_{j_m}^{(m)}(\boldsymbol{x})\right)w_{i_l,i_m}^{(m+1)} & \text{if } m = l-1 \\[3mm]
\left(\dfrac{\sigma_w}{\sqrt{n}}\right)^{l-m+1}\tau''\left(f_{i_l}^{(l+1)}(\boldsymbol{x})\right)\tau'\left(f_{i_m}^{(m+1)}(\boldsymbol{x})\right)\tau\left(f_{j_m}^{(m)}(\boldsymbol{x})\right)\times \\[3mm]
\quad\times\displaystyle\sum_{k_l,\dots,k_{m+2}=1}^{n}w_{i_l,k_l}^{(l)}\tau'\left(f_{k_l}^{(l)}(\boldsymbol{x})\right)\left(\prod_{s=m+2}^{l-1}w_{k_{s+1},k_s}^{(s)}\tau'\left(f_{k_s}^{(s)}(\boldsymbol{x})\right)\right)w_{k_{m+2},i_m}^{(m+1)} & \text{if } m < l-1
\end{cases}$$

and

$$\frac{\partial}{\partial w_{i_m,j_m}^{(m)}}\left[\tau\left(f_{j_l}^{(l)}(\boldsymbol{x})\right)\right]$$

$$=\begin{cases}
0 & \text{if } m \geq l \\[2mm]
\dfrac{\sigma_w}{\sqrt{n}}\delta_{j_l i_m}\tau'\left(f_{j_l}^{(l)}(\boldsymbol{x})\right)\tau\left(f_{j_m}^{(m)}(\boldsymbol{x})\right) & \text{if } m = l-1 \\[3mm]
\left(\dfrac{\sigma_w}{\sqrt{n}}\right)^2\tau'\left(f_{j_l}^{(l)}(\boldsymbol{x})\right)\tau'\left(f_{i_m}^{(m+1)}(\boldsymbol{x})\right)\tau\left(f_{j_m}^{(m)}(\boldsymbol{x})\right)w_{j_l,i_m}^{(m+1)} & \text{if } m = l-2 \\[3mm]
\left(\dfrac{\sigma_w}{\sqrt{n}}\right)^{l-m}\tau'\left(f_{j_l}^{(l)}(\boldsymbol{x})\right)\tau'\left(f_{i_m}^{(m+1)}(\boldsymbol{x})\right)\tau\left(f_{j_m}^{(m)}(\boldsymbol{x})\right)\times \\[3mm]
\quad\times\displaystyle\sum_{k_{l-1},\dots,k_{m+2}=1}^{n}w_{j_l,k_{l-1}}^{(l-1)}\tau'\left(f_{k_{l-1}}^{(l)}(\boldsymbol{x})\right)\left(\prod_{s=m+2}^{l-2}w_{k_{s+1},k_s}^{(s)}\tau'\left(f_{k_s}^{(s)}(\boldsymbol{x})\right)\right)w_{k_{m+2},i_m}^{(m+1)} & \text{if } m < l-2.
\end{cases}$$

## Appendix F. Proof of Theorems 8 and Theorem 9

We will write $a \lesssim b$ if there exists a universal constant $C$ such that $a \leq Cb$, and $a \asymp b$ if both $a \lesssim b$ and $b \lesssim a$. Both the proofs are essentially based on Theorem 7, adapted to the case $L = 2$, and $p = 1$ and $p \geq 2$ respectively. As outlined in the main body, after stating Theorem 7, the biggest problem one has to face lies in the fact that there is not a straightforward way of controlling the expectations in the bound, since each node depends on the nodes of all the previous layers in a very convoluted manner. Nonetheless, it is still possible to overcome this problem in this case by conditioning on the previous hidden layer, since $f^{(1)}(\boldsymbol{x})$ is normally distributed. We will show how to do this for a specific term in the bound, as for the others the same methodology can be applied. To simplify the notation, we will write $f_i^{(2)}(\boldsymbol{x}) := f_i^{(2)}(\boldsymbol{x}, n)$.

Without loss of generality, we can assume $\gamma > 1$.

We will make use several times of the following generalized Bahr-Esseen inequalities (Dharmadhikari and Jogdeo (1969)): if $X_1, \ldots, X_n$ are independent, zero mean random variables with finite $r$-th moment, for some $r > 2$, then

$$\mathbb{E}\left[\left|\sum_{k=1}^{n} X_k\right|^r\right] \leq cn^{r/2-1} \sum_{k=1}^{n} \mathbb{E}[|X_k|^r]$$

where $c > 0$ is a constant that depends only on $r$.

First, notice that, for every $r > 2$, $\mathbb{E}[|f_i^{(1)}(\boldsymbol{x})|^r]$ is bounded by a constant that only depends on $r$ and $\boldsymbol{x}$. Moreover, for every $r > 0$,

$$\begin{aligned}
\mathbb{E}\left[|\tau(f_i^{(2)}(\boldsymbol{x})|^r \mid Y_.\right] &\leq \mathbb{E}\left[(\alpha + \beta|f_i^{(2)}(\boldsymbol{x})|)^r \mid Y_.\right] \\
&\leq 2^r\left(\alpha^r + \beta^r \mathbb{E}[|f_i^{(2)}(\boldsymbol{x})|^r \mid Y_.]\right) \\
&\leq 2^r\alpha^r + 2^r\beta^r \sigma_w^r n^{-r/2} \mathbb{E}\left[|\sum_{j=1}^{n} w_{i,j}^{(1)} \tau(f_j^{(1)}(\boldsymbol{x}))|^r \mid Y_.\right] \\
&\leq 2^r\alpha^r + 2^r\beta^r \sigma_w^r n^{-1} \sum_{j=1}^{n} \mathbb{E}\left[|w_{i,j}^{(1)} \tau(f_j^{(1)}(\boldsymbol{x}))|^r \mid Y_.\right] \\
&\leq 2^r\alpha^r + 2^r\beta^r \sigma_w^r \mathbb{E}[|Z|^r] n^{-1} \sum_{j=1}^{n} |\tau(f_j^{(1)}(\boldsymbol{x}))|^r,
\end{aligned}$$

where $Z \sim \mathcal{N}(0, 1)$ and we have used the generalized Bahr-Esseen inequality and the fact that the random variables $w_{i,j}^{(1)} \tau(f_j^{(1)}(\boldsymbol{x}))$ are conditionally independent, given $Y_.$, with zero conditional expectations. The same equations apply to $|\tau'(f_i^{(2)}(\boldsymbol{x}))|$. It follows that, for every $r > 0$ there exists a $c_r$ not depending on $n$ such that

$$\max\left(\mathbb{E}\left[|\tau(f_i^{(2)}(\boldsymbol{x})|^r\right], \mathbb{E}\left[|\tau'(f_i^{(2)}(\boldsymbol{x})|^r\right]\right) \leq c_r.$$

We will now show how to bound $\sum_{i,j=1}^{n} \left\{\mathbb{E}\left[\left(\frac{\partial F}{\partial w_i}\frac{\partial F}{\partial w_j}\right)^2\right]\mathbb{E}\left[\left\langle\nabla_{w_i,.}^2 F, \nabla_{w_j,.}^2 F\right\rangle^2\right]\right\}^{1/2}$ from above. If $i \neq j$, then

$$\mathbb{E}\left[\left(\frac{\partial F}{\partial w_i}\frac{\partial F}{\partial w_j}\right)^2\right] = (\sigma_w n^{-1/2})^4 \mathbb{E}\left[\mathbb{E}\left[\tau^2(f_i^{(2)}(x))|Y_.\right]^2\right] \leq \sigma_w^4 n^{-2} \mathbb{E}\left[\tau^4(f_i^{(2)}(x))\right] \lesssim n^{-2}.$$

For $i = j$, we can write that

$$\mathbb{E}\left[\left(\frac{\partial F}{\partial w_i}\right)^4\right] \leq \sigma_w^4 n^{-2} \mathbb{E}\left[\tau^4(f_i^{(2)}(x))\right] \lesssim n^{-2}.$$

Let us now turn to the expectation involving the Hessian. We have that

$$\left\langle\nabla_{w_i,.}^2 F, \nabla_{w_j,.}^2 F\right\rangle = \left(\frac{\sigma_w}{\sqrt{n}}\right)^4 \tau'\left(f_i^{(2)}(\boldsymbol{x})\right) \tau'\left(f_j^{(2)}(\boldsymbol{x})\right)\left(\delta_{ij}\sum_{b=1}^{n}\tau\left(f_b^{(1)}(\boldsymbol{x})\right)^2\right.$$

$$+\Gamma^2 \sum_{b=1}^{n} w_{i,b}^{(1)} w_{j,b}^{(1)} \tau' \left( f_b^{(1)}(\boldsymbol{x}) \right)^2 \Bigg),$$

so that

$$\left\langle \nabla_{w_i,\cdot}^2 F, \nabla_{w_j,\cdot}^2 F \right\rangle^2 \leq \left( \frac{\sigma_w}{\sqrt{n}} \right)^8 \tau' \left( f_i^{(2)}(\boldsymbol{x}) \right)^2 \tau' \left( f_j^{(2)}(\boldsymbol{x}) \right)^2 \times$$

$$\times \left( 2\delta_{ij} \left( \sum_{b=1}^{n} \tau \left( f_b^{(1)}(\boldsymbol{x}) \right)^2 \right)^2 + 2\Gamma^4 \left( \sum_{b=1}^{n} w_{i,b}^{(1)} w_{j,b}^{(1)} \tau' \left( f_b^{(1)}(\boldsymbol{x}) \right)^2 \right)^2 \right)$$

$$\lesssim n^{-4} \delta_{ij} \tau' \left( f_i^{(2)}(\boldsymbol{x}) \right)^2 \tau' \left( f_j^{(2)}(\boldsymbol{x}) \right)^2 \left( \sum_{b=1}^{n} \tau \left( f_b^{(1)}(\boldsymbol{x}) \right)^2 \right)^2$$

$$+ n^{-4} \tau' \left( f_i^{(2)}(\boldsymbol{x}) \right)^2 \tau' \left( f_j^{(2)}(\boldsymbol{x}) \right)^2 \left( \sum_{b=1}^{n} w_{i,b}^{(1)} w_{j,b}^{(1)} \tau' \left( f_b^{(1)}(\boldsymbol{x}) \right)^2 \right)^2.$$

We will bound the expectations of the two terms of the sum separately. For the first term we have

$$\mathbb{E} \left[ n^{-4} \delta_{ij} \tau' \left( f_i^{(2)}(\boldsymbol{x}) \right)^2 \tau' \left( f_j^{(2)}(\boldsymbol{x}) \right)^2 \left( \sum_{b=1}^{n} \tau \left( f_b^{(1)}(\boldsymbol{x}) \right)^2 \right)^2 \right]$$

$$\leq n^{-4} \delta_{ij} \mathbb{E} \left[ \left( \sum_{b=1}^{n} \tau \left( f_b^{(1)}(\boldsymbol{x}) \right)^2 \right)^2 \mathbb{E} \left[ \tau' \left( f_i^{(2)}(\boldsymbol{x}) \right)^2 | Y_\cdot \right]^2 \right]$$

$$\leq n^{-4} \delta_{ij} \mathbb{E} \left[ \left( \sum_{b=1}^{n} \tau \left( f_b^{(1)}(\boldsymbol{x}) \right)^2 \right)^2 \mathbb{E} \left[ \tau' \left( f_i^{(2)}(\boldsymbol{x}) \right)^4 | Y_\cdot \right] \right]$$

$$\leq n^{-4} \delta_{ij} \left( \mathbb{E} \left[ \left( \sum_{b=1}^{n} \tau \left( f_b^{(1)}(\boldsymbol{x}) \right)^2 \right)^4 \right] \right)^{1/2} \left( \mathbb{E} \left[ \tau' \left( f_i^{(2)}(\boldsymbol{x}) \right)^8 \right] \right)^{1/2}$$

$$\lesssim n^{-2} \delta_{ij},$$

For the second term, we consider the cases $i = j$ and $i \neq j$ separately. For $i = j$ we can write that

$$\mathbb{E} \left[ n^{-4} \delta_{ij} \tau' \left( f_i^{(2)}(\boldsymbol{x}) \right)^4 \left( \sum_{b=1}^{n} (w_{i,b}^{(1)})^2 \tau' \left( f_b^{(1)}(\boldsymbol{x}) \right)^2 \right)^2 \right]$$

$$\leq n^{-4} \delta_{ij} c_8^{1/2} \left( \mathbb{E} \left[ \left( \sum_{b=1}^{n} (w_{i,b}^{(1)})^2 \tau' \left( f_b^{(1)}(\boldsymbol{x}) \right)^2 \right)^4 \right] \right)^{1/2}$$

$$\lesssim \delta_{ij} n^{-2}.$$

On the other hand, for $i \neq j$ we can write that

$$
\mathbb{E}\left[n^{-4}\tau'\left(f_i^{(2)}(\boldsymbol{x})\right)^2\tau'\left(f_j^{(2)}(\boldsymbol{x})\right)^2\left(\sum_{b=1}^n w_{i,b}^{(1)}w_{j,b}^{(1)}\tau'\left(f_b^{(1)}(\boldsymbol{x})\right)^2\right)^2\right]
$$

$$
\leq n^{-4}\left(\mathbb{E}\left[\tau'\left(f_i^{(2)}(\boldsymbol{x})\right)^4\tau'\left(f_j^{(2)}(\boldsymbol{x})\right)^4\right]\right)^{1/2}\left(\mathbb{E}\left[\left(\sum_{b=1}^n w_{i,b}^{(1)}w_{j,b}^{(1)}\tau'\left(f_b^{(1)}(\boldsymbol{x})\right)^2\right)^4\right]\right)^{1/2}
$$

$$
\leq n^{-4}c_8^{1/2}\left(\mathbb{E}\left[\mathbb{E}\left[\left(\sum_{b=1}^n w_{i,b}^{(1)}w_{j,b}^{(1)}\tau'\left(f_b^{(1)}(\boldsymbol{x})\right)^2\right)^4\mid Y.\right]\right]\right)^{1/2}
$$

$$
\leq n^{-4}c_8^{1/2}\left(\mathbb{E}\left[n\sum_{b=1}^n\mathbb{E}\left[|w_{i,b}^{(1)}w_{j,b}^{(1)}|^4|\tau'\left(f_b^{(1)}(\boldsymbol{x})\right)|^8\mid Y.\right]\right]\right)^{1/2}
$$

$$
\leq n^{-4}c_8^{1/2}\left(\mathbb{E}\left[n\sum_{b=1}^n|\tau'\left(f_b^{(1)}(\boldsymbol{x})\right)|^8\mathbb{E}\left[|w_{i,b}^{(1)}w_{j,b}^{(1)}|^4\mid Y.\right]\right]\right)^{1/2}
$$

$$
\leq n^{-4}c_8^{1/2}E\left[|Z|^4\right]\left(n\mathbb{E}\left[\sum_{b=1}^n|\tau'\left(f_b^{(1)}(\boldsymbol{x})\right)|^8\right]\right)^{1/2}
$$

$$
\lesssim n^{-3}
$$

Summarizing, we can write that

$$
\sum_{i,j=1}^n\left\{\mathbb{E}\left[\left(\frac{\partial F}{\partial w_i}\frac{\partial F}{\partial w_j}\right)^2\right]\mathbb{E}\left[\left\langle\nabla_{w_i,\cdot}^2 F,\nabla_{w_j,\cdot}^2 F\right\rangle^2\right]\right\}^{1/2}\lesssim\sum_{i,j=1}^n\{n^{-2}(\delta_{ij}n^{-2}+n^{-3})\}^{1/2}\lesssim n^{-1/2}.
$$

The same rate can be found with analogous steps for all the other terms in the sum given by Theorem 7, and taking the square root one more time gives the rate of $n^{-1/4}$. The proof in the case of $p$ output is essentially the same, apart from the fact that we have an extra sum over $p$ index, which leads to the rate $\mathcal{O}(\sqrt{p/\sqrt{n}})$.

As stated in the main body, this rate is worse than the one in Basteri and Trevisan (2022), but in order to show that this in not "our fault", but it is due to the intrinsic behaviour of these Poincaré inequality in this setting, we will now show that the same rate $n^{-1/4}$ is obtained in the case $\tau = id$, the identity function, which is arguably the nicest setting possible. Indeed, if we consider the NN

$$
F := n^{-1}\sum_{i=1}^n\sum_{j=1}^n w_i w_{i,j}^{(1)}Y_j,
$$

we can compute explicitly

$$
\mathbb{E}\left[\left(\frac{\partial F}{\partial w_i}\frac{\partial F}{\partial w_j}\right)^2\right]\quad\text{and}\quad\mathbb{E}\left[\left\langle\nabla_{w_i,\cdot}^2 F,\nabla_{w_j,\cdot}^2 F\right\rangle^2\right],
$$

and see that they lead to the same suboptimal rate of $n^{-1/4}$. As for the first term, we have

$$\mathbb{E}\left[\left(\frac{\partial F}{\partial w_i}\frac{\partial F}{\partial w_j}\right)^2\right] = n^{-2}\mathbb{E}\left[\mathbb{E}\left[(f_i^{(2)}(x))^2|Y.\right]^2\right],$$

and

$$\mathbb{E}\left[(f_i^{(2)}(x))^2|Y.\right] = n^{-1}\mathbb{E}\left[\left(\sum_{j=1}^n w_{i,j}^{(1)}Y_j\right)^2\bigg|Y.\right] = n^{-1}\mathbb{E}\left[\sum_{j,k=1}^n w_{i,j}^{(1)}w_{i,k}^{(1)}Y_jY_k\bigg|Y.\right] = n^{-1}\sum_{j=1}^n Y_j^2,$$

so that

$$\mathbb{E}\left[\left(\frac{\partial F}{\partial w_i}\frac{\partial F}{\partial w_j}\right)^2\right] = n^{-4}\mathbb{E}\left[\left(\sum_{j=1}^n Y_j^2\right)^2\right] = n^{-4}\sum_{j,k=1}^n \mathbb{E}\left[Y_j^2Y_k^2\right] \asymp n^{-2}.$$

As for the second term,

$$\mathbb{E}\left[\left\langle\nabla_{w_i,\cdot}^2 F, \nabla_{w_j,\cdot}^2 F\right\rangle\right] \asymp n^{-4}\delta_{ij}\mathbb{E}\left[\left(\sum_{b=1}^n Y_b^2\right)^2\right] + n^{-4}\mathbb{E}\left[\left(\sum_{b=1}^n w_{i,b}^{(1)}w_{j,b}^{(1)}\right)^2\right]$$

$$= n^{-4}\delta_{ij}(2n + n^2) + n^{-4}\mathbb{E}\left[\left(\sum_{b=1}^n w_{i,b}^{(1)}w_{j,b}^{(1)}\right)^2\right],$$

since $\sum_{b=1}^n Y_b^2 \sim \chi_n^2$, and $\mathbb{E}[\chi_n^2] = n$ and $\mathrm{Var}[\chi_n^2] = 2n$. Also,

$$\mathbb{E}\left[\left(\sum_{b=1}^n w_{i,b}^{(1)}w_{j,b}^{(1)}\right)^2\right] = \mathbb{E}\left[\sum_{a,b=1}^n w_{i,b}^{(1)}w_{j,b}^{(1)}w_{i,a}^{(1)}w_{j,a}^{(1)}\right] = \sum_{a,b=1}^n \mathbb{E}\left[w_{i,b}^{(1)}w_{j,b}^{(1)}w_{i,a}^{(1)}w_{j,a}^{(1)}\right]$$

$$\lesssim \sum_{a,b=1}^n \left[\delta_{ij} + (1 - \delta_{ij})\delta_{ab}\right] = n^2\delta_{ij} + n(1 - \delta_{ij}),$$

hence

$$\mathbb{E}\left[\left\langle\nabla_{w_i,\cdot}^2 F, \nabla_{w_j,\cdot}^2 F\right\rangle^2\right] \lesssim n^{-4}\left[\delta_{ij}(2n + n^2) + n^2\delta_{ij} + n(1 - \delta_{ij})\right] \lesssim n^{-2}\delta_{ij} + (1 - \delta_{ij})n^{-3}.$$

Combining the two terms we get something of the order $n^{-4}\delta_{ij} + (1 - \delta_{ij})n^{-5}$, and after taking the square root, something like $\sqrt{n^{-4}\delta_{ij} + (1 - \delta_{ij})n^{-5}} \lesssim n^{-2}\delta_{ij} + (1 - \delta_{ij})n^{-5/2}$. The same is true for all the others terms which appear in the bound of Theorem 7, hence, summing over all $i, j \in [n]$, gives a rate whose leading term is again of the order $n^{-1/4}$.

## Appendix G. Numerical illustrations

We present a simulation study with respect to two choices of the activation function $\tau$: i) $\tau(x) = \tanh x$, which is polynomially bounded with parameters $\alpha = 1$ and $\beta = 0$; ii) $\tau(x) = x^3$, which is polynomially bounded with parameters $\alpha = 6$, $\beta = 1$ and $\gamma = 3$. Each of the plots below is obtained as follows: for a fixed width of $n = k^3$, with $k \in \{1, \cdots, 16\}$, we simulate 5000 points from a single-layer NN as in Theorem 4 to produce an estimate of the distance between the NN and a Gaussian random variable with mean 0 and variance $\sigma^2$, which is estimated by means of a Monte-Carlo approach. Estimates of the KS and TV distance are produced by means of the functions `KolmogorovDist` and `TotVarDist` from the package **distrEx** by Ruckdeschel et al. (2006) while those of the 1-Wasserstein distance using the function `wasserstein1d` from the package **transport** by Schuhmacher et al. (2022). We repeat this procedure 2000 times for every fixed $n \in \{3, 6, \cdots, 51\}$, compute the sample mean (blue dots), and compare these estimates with the theoretical explicit bound given by Theorem 4 (green dots), and with the implicit bound given by Theorem 7 (red dots).

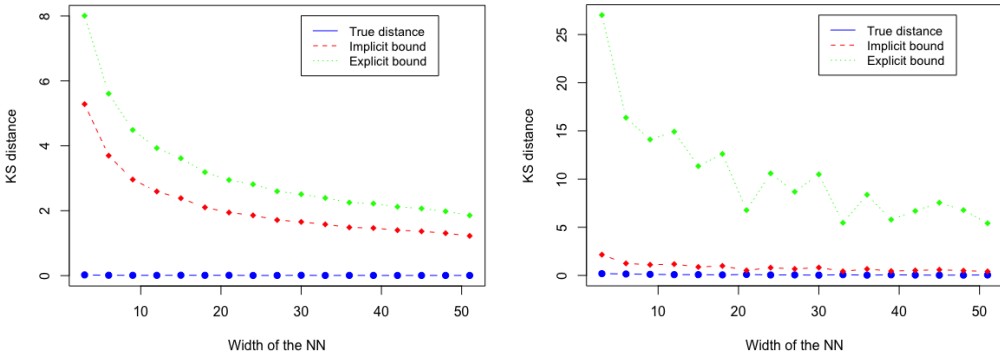

Figure 1: Estimates of the Kolmogorov-Smirnov distance for a Shallow NN of varying width $n \in \{3, 6, \cdots, 51\}$, with $\tau(x) = \tanh x$ (left) and $\tau(x) = x^3$ (right).

All the figures confirm that the distance between a shallow NN and an arbitrary Gaussian random variable, with the same mean and variance, is $\lesssim n^{-1/2}$, with approximation errors improving as the width $n \to \infty$. The evaluation of the implicit bound of Theorem 7 results in much tighter estimate of the distance than what provided by the explicit bound, which highlight the rate $n^{-1/2}$ at the cost of having a looser constant. This is clear in the case $\tau(x) = x^3$, where the polynomial envelope assumption leads to a much rougher bound to the one you may get computing the derivatives explicitly.

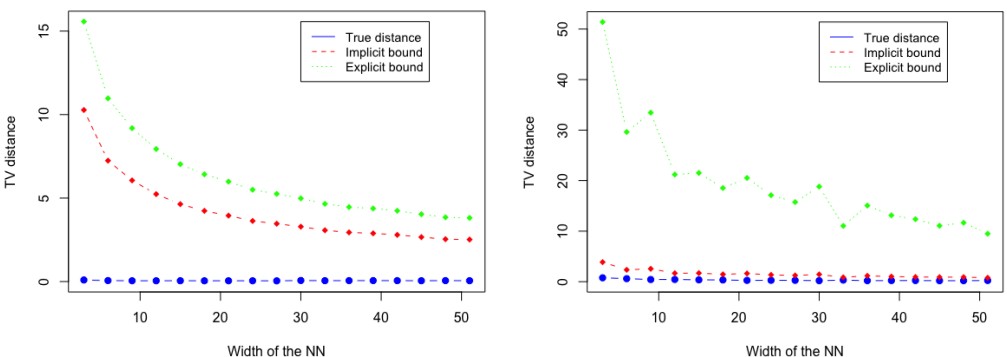

Figure 2: Estimates of the Total Variation distance for a Shallow NN of varying width $n \in \{3, 6, \cdots, 51\}$, with $\tau(x) = \tanh x$ (left) and $\tau(x) = x^3$ (right).

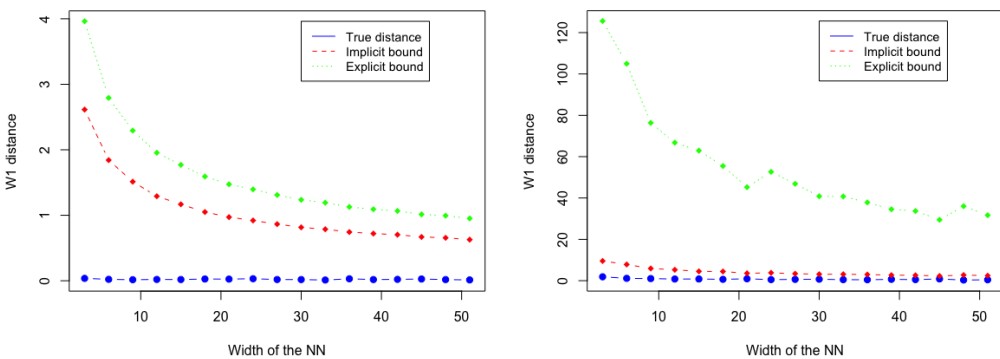

Figure 3: Estimates of the 1-Wasserstein distance for a Shallow NN of varying width $n \in \{3, 6, \cdots, 51\}$, with $\tau(x) = \tanh x$ (left) and $\tau(x) = x^3$ (right).

