# OpenReview forum: "Non-asymptotic approximations of Gaussian neural networks via second-order Poincar\'e inequalities"
_approximateinference.org/AABI/2024/Symposium_Archival_Track — AABI 2024 - Archival Track_

### Official Review · Reviewer_mi2r · 2024-04-19
**Intriguing promises!**

**Rating:** 7
**Confidence:** 2

**Review:**

### **Overview**
Consider any fixed collection of d-dimensional inputs x_1,x_2,...x_p and any densely-connected feed-forward neural network f with normally distributed biases and weights, suitably scaled to keep the variance of f(x) in check.  It is well known that the outputs of the network, F=(f(x_1),f(x_2)...f(x_p)), are well-approximated as a multivariate Gaussian distribution as long as the number of hidden units is large.  This paper investigates new techniques for quantifying the extent to which F is well-approximated as Gaussian.  Specifically, letting n denote the number of hidden units at each layer, this paper tries to get upper bounds for how different F is from a (multivariate) Gaussian, and show that these upper bounds decay quickly in n.  The paper investigate three metrics for difference (W1, TV, and KS).

The paper, by itself, does not develop any new bounds for how quickly F approaches a suitable gaussian.  However, it does develop a new proof technique.  In theory, it seems possible that this new technique could shed new light on what's going on and lead to new results in the future.

### **Soundness**
I believe the structure of the proof technique is sound, although I did not check carefully.  Starting around equation 10, which is intended as a restatement of Vidotto 2020, I started to give up hope of fully getting to the bottom of what was going on. For example, I think the Vidotto version has a double integral and a double sum instead of a quadruple sum, not sure why.  And the integral was with respect to "a positive, σ-finite and non-atomic measure."

At any rate, it is certainly plausible that (1) if the mapping from neural net parameters (which, a bit confusingly, are called X) into outputs has small Hessians and Jacobians, then the outputs will indeed be approximately normal and that (2) with a large number of hidden units, the 1/sqrt(n) term in large-width networks will ensure that all the Hessians and Jacobians suitably small.   So I think, overall, this approach is viable.

### **Significance**

It is hard to say.  Just at a glance, the proofs do not look "simpler" than those found in, say, the cited work by Basteri and Trevisan (2022) (BTW preprint is now actually from 2023, they updated it).  Yet, on the other hand, the **gist** of this new proof does appear simpler (i.e., small gradients and hessians due to 1/sqrt(n) scaling + second order Poincare inequality = victory).  So there may be something to the author's claim that the proof may be in some respects simpler.

To feel sure, it would be good to see something new proved with the technique.  For example, convolutional layers were mentioned as a possible application of the new method.  If the authors could pull that off elegantly, that would be compelling.  Or if there was an elegant way to show something for L>>2.  As it stands, it seems the challenges of computing all the Hessians are a lot of trouble.  Perhaps a computer-assisted proof technique could be helpful?  Somehow it does seem that the authors have reduced a deeply confusing problem to a problem that is only confusing because it has a lot of moving pieces.  So perhaps computers can handle all the moving pieces.  Not sure.

### **Clarity**

Clarity could use some work.

From the very first line, I was surprised to find the input to the neural net being a matrix (which I now realize is because we want to evaluate the same neural network at many inputs).  I was also surprised to find that the letter used for indexing was supposed to be a clue as to the semantic meaning of the subscript for a particular variable ($x_j$ for rows and $x_u$ for columns).

In the middle, I continued to find some puzzling pieces.  In some cases ||.||_op was used but in other cases ||.||_2 was used (I assume same thing was meant both time?  Unless there is a distinction between the operator norm and the spectral norm... but we're talking about finite-dimensional vector spaces here right?  Not Banach spaces?).

Puzzling pieces were also to be found up to the very end.  "This fact indicates the impossibility to apply the results of Vidotto (2020) in the context of continuous activation functions as the ReLU function, and the necessity to come up with new results on second-order Poincare  inequalities to fill this gap" was also a bit confusing.  The trouble with ReLU does not seem to be its continuity, but rather its lack of continuous derivatives, right?  And at the very last line, that remark about f''(x)-xf'(x) = h(x)-Eh(Z).  I was not at all sure how that fit into the big picture.

But none of these things are so important.  They could all be fixed with several rounds of revision.

### **Next steps**

One viable path forward would be to use the methodology to prove something new.  I think that would really make this into a strong contribution.  Thanks for your hard work!

---

### Official Review · Reviewer_zses · 2024-04-25

**Rating:** 5
**Confidence:** 2

**Review:**

I am not familiar with this field and I can only provide high-level reviews. Therefore, my understanding could be not accurate.

This paper shows that the second-order Poincare inequality can be used to establish quantitative central limit theorems which are used to show a Gaussian NN's output converges to a GP with infinite widths.

In my understanding, I know that previous methods justifying NNs $\to$ GPs used induction layer by layer. Instead, this paper used an alternative directly utilizing outputs' gradients and Hessians. This new proof is technically sound. However, the benefits of such new methods are not clear. The author claimed this new method can be applied to NNs with other architecture, for example, CNNs. There are no follow-up sections in the paper for this claim. The two examples showing shallow NNs and deep NNs $\to$ GPs are known, though the proof methods are new.



Typos:
1.  let d_tv bu $\to$ let d_tv be

---

### Official Review · Reviewer_Wmf1 · 2024-04-26
**Reviews for "On second-order Poincare inequalities in non-asymptotic approximations of Gaussian neural networks"**

**Rating:** 7
**Confidence:** 2

**Review:**

The paper studies the quantitative central limit theorem (QCLT) of Gaussian neural networks through the second-order Poincare inequalities, which rely only on the fact that the neural network is a functional of a Gaussian process. This helps reduce the problem of establishing QCLT to computing the gradient and Hessian of the neural network outputs.

Overall, the paper is well-written with solid mathematical proofs. However, I do have some concerns about the potential impact of the paper:

1. During my literature review when reading the paper, I found a very similar paper published on Arxiv last year (https://arxiv.org/pdf/2304.04010). If these two papers are the same, then there is no concern anymore.

2. While reducing the problem of establishing QCLT to computing the gradient and Hessian of the neural network outputs does generalizes the theoretical framework, it is still unclear to me how it can be applied to other neural network architectures, such as CNN and RNN, let alone attention or transformer. The authors may need to clarify this point in their paper.

3. For the deep Gaussian neural network of depth $L\geq 1$, what are the challenges of proving the conjecture $\mathcal{O}(L\sqrt{p/\sqrt{n}})$?

Some small typos:

1. In the fifth line of the paragraph before "Section 1.2" on Page 3, you missed parentheses after "i.e." for Eq.(6).

2. In the seventh line of the paragraph after Theorem 4 on Page 6, "As for the constant is (12),...". The "is" should be "in".

---

### Meta-Review · Area_Chair_bhSC · 2024-05-20

**Recommendation:** Accept (Poster)
**Confidence:** 3

**Metareview:**

The paper explores the application of second-order Poincare inequalities to establish quantitative central limit theorems (QCLTs) for Gaussian neural networks (NNs). The paper takes a different approach than prior work by reducing the problem to computing the gradient and Hessian of the NN outputs and potentially enables broader applications beyond fully-connected feed-forward architectures.

While the clarity can be improved and the motivation presented more clearly, the paper is of high quality and presents solid mathematical proofs. The work is original as it introduces a new proof technique to the field that might enable application to new NN architecture albeit with potentially worse rates. Ultimately, it is therefore hard to judge the significance. However, all reviewers agree that there is a good chance that the results are novel and having an original alternative to existing works seems valuable.

I recommend that the authors improve some clarity issues brought up by the reviewers and better discuss the scope and applicability for other neural network architectures as discussed in the rebuttal.

---

### Decision · Program_Chairs · 2024-05-27

Accept